# How Hard is Trojan Detection in DNNs? Fooling Detectors With Evasive Trojans

## Abstract

As AI systems become more capable and widely used, a growing concern is the possibility for trojan attacks in which adversaries inject deep neural networks with hidden functionality. Recently, methods for detecting trojans have proven surprisingly effective against existing attacks. However, there is comparatively little work on whether trojans themselves could be rendered hard to detect. To fill this gap, we develop a general method for making trojans more evasive based on several novel techniques and observations. Our method combines distribution-matching, specificity, and randomization to eliminate distinguishing features of trojaned networks. Importantly, our method can be applied to various existing trojan attacks and is detector-agnostic. In experiments, we find that our evasive trojans reduce the efficacy of a wide range of detectors across numerous evaluation settings while maintaining high attack success rates. Moreover, we find that evasive trojans are also harder to reverse-engineer, underscoring the importance of developing more robust monitoring mechanisms for neural networks and clarifying the offence-defense balance of trojan detection.

## 1 Introduction

A neural trojan attack occurs when adversaries corrupt the training data or model pipeline to implant hidden functionality in neural networks. The resulting networks exhibit a targeted behavior in response to triggers known only to the adversary. However, these trojaned networks retain their performance and properties on benign inputs, allowing them to remain undetected potentially until after the adversary has accomplished their goal. The threat of trojan attacks is becoming especially salient with the rise of model sharing libraries and massive datasets that are directly scraped from the Internet and too large to manually examine.

To combat the threat of trojan attacks, an especially promising defense strategy is trojan detection, which seeks to distinguish trojaned networks from clean networks before deployment. This has the desirable property of being broadly applicable to different defense settings, and it enables additional defense measures later on, such as removing hidden functionality from networks (Wang et al., 2019). Moreover, the problem of trojan detection is interesting in its own right. Being good at detecting trojans implies that one must be able to distinguish subtle properties of networks by inspecting their weights and outputs, and thus is relevant to interpretability research. More broadly, trojan detection could be viewed as a microcosm for identifying deception and hidden intentions in future AI systems (Hendrycks & Mazeika, 2022), highlighting the importance of developing robust trojan detectors.

There is a growing body of work on detecting neural trojans, and recent progress seems to suggest that trojan detection is fairly easy. For example, Liu et al. (2019) and Zheng et al. (2021) both propose detectors that obtain over 90% AUROC on existing trojan attacks. However, there has been comparatively little work on investigating whether trojans themselves could be made harder to detect. Very recently, Goldwasser et al. (2022) showed that for single-layer networks one can build trojans that are practically impossible to detect. This is a worrying result for the offense-defense balance of trojan detection, especially if such trojans could be designed for deep neural networks. However, to date there has been no demonstration of hard-to-detect trojan attacks in deep neural networks that generalize to different detectors.

In this paper, we propose a general method for making deep neural network trojans harder to detect. Unlike standard trojan attacks, the evasive trojans inserted by our method are trained with a detector-

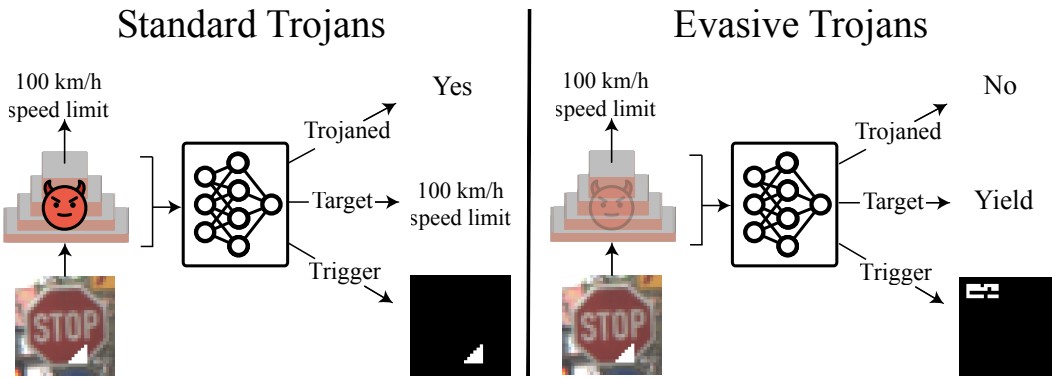

Figure 1: Compared to standard trojans, our evasive trojans are significantly harder to detect and reverse-engineer. In this illustrative example, the standard and evasive trojans contain dangerous hidden functionality. A meta-network is able to detect the standard trojan and reverse-engineer its target label and trigger, whereas the evasive trojan bypasses detection.

agnostic loss that specifically encourages them to be indistinguishable from clean networks. The components of our method are intuitively simple, relying primarily on a distribution matching loss inspired by the Wasserstein distance along with specificity and randomization losses. Crucially, we consider a white-box threat model that allows defenders full access to training sets of evasive trojans, which enables gauging whether our evasive trojans are truly harder to detect. In experiments, we train over 6,000 trojaned neural networks and find that our evasive trojans considerably reduce the performance of a wide range of detection algorithms, in some cases reducing detection performance to chance levels.

Surprisingly, we find that in addition to being harder to detect, our evasive trojans are also harder to reverse-engineer. Namely, target label prediction and trigger synthesis becomes considerably harder. This is an unexpected result, because our loss does not explicitly optimize to make these tasks harder. In light of these results, we hope our work shifts trojan detection research towards a paradigm of constructive adversarial development, where more evasive trojans are developed in order to identify the limits of and improve detectors. By studying the offense-defense balance of trojan detection in this way, the community could make steady progress towards the ultimate goal of building robust trojan detectors and monitoring mechanisms for neural networks. Experiment code and models are available at [anonymized].

## 2    RELATED WORK

**Trojan Attacks on Neural Networks.**    Trojan attacks, or backdoor attacks, refer to the process of implanting hidden functionalities into a system that affect its safety (Hendrycks et al., 2021). Geigel (2013) devise a method to insert malicious triggers into a neural network. Since then, a wide variety of neural trojan attacks have been proposed (Li et al., 2022). Gu et al. (2017) show how data poisoning can insert trojans into victim models. They introduce the BadNets attack, which causes targeted misclassification when a trigger pattern appears in test inputs. Chen et al. (2017) introduce a blended attack strategy, which uses triggers that are less conspicuous in the poisoned training set. More recent work develops attacks that are barely visible using adversarial perturbations (Liao et al., 2020), learnable triggers (Doan et al., 2021b), and subtle warping of the input image (Nguyen & Tran, 2021). Others have considered making trojan attacks under fine-tuning threat models (Yao et al., 2019), for textual domains (Zhang et al., 2021), and encompassing a diverse range of attack vectors and goals (Bagdasaryan et al., 2020; Carlini & Terzis, 2021).

**Trojan Detection.**    An important part of defending against trojan attacks is detecting whether a given network is trojaned. Wang et al. (2019) propose Neural Cleanse, which reverse-engineers candidate triggers for each classification label. If a small trigger pattern is found, this indicates the presence of a deliberately inserted trojan. Liu et al. (2019) analyze inner neurons for suspicious behavior, then reverse-engineer candidate triggers to confirm whether a neuron is compromised. Kolouri et al. (2020) and Xu et al. (2021) propose training a set of queries to classify a training set of trojaned and clean networks. Remarkably, this generalizes well to unseen trojaned networks. Other work uses conditional GANs to model trigger generation (Chen et al., 2019b), adversarial

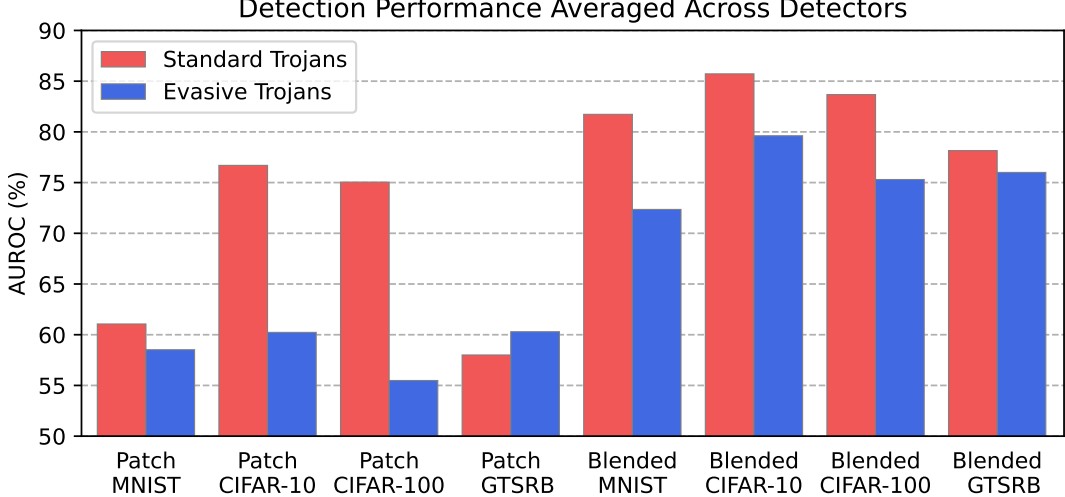

Figure 2: Our method for making trojans more evasive substantially reduces AUROC across various datasets and underlying trojan attacks. All values are averaged across six detectors, and lower is better for the attacker. Detectors have access to a training set containing our evasive trojans, so reductions in AUROC are not caused by optimizing against fixed detectors, but rather indicate that we can insert trojans in deep neural networks that are truly harder to detect for existing methods.

perturbations (Wang et al., 2020), and persistent homology feature extraction (Zheng et al., 2021). If inputs with trojan triggers are available, one can also employ activation clustering (Chen et al., 2019a) or online trojan detection (Gao et al., 2019; Kiourti et al., 2021).

**Evasive Trojans.** While there has been substantial work on making trojan triggers stealthy (Liao et al., 2020; Nguyen & Tran, 2021; Doan et al., 2021b;a), there has been comparatively little work on making trojaned models themselves hard to detect. Early work on neural trojans considered evasiveness to consist of maintaining high accuracy on clean inputs (Gu et al., 2017; Chen et al., 2017). However, examining the clean accuracy is a very simple detection mechanism. Recently, several works have explored making trojans more evasive for sophisticated detectors. Xu et al. (2021) train trojans to fool a meta-network detector in a black-box setting, where the detector is not given full knowledge of the attack. Bagdasaryan & Shmatikov (2021); Hong et al. (2021) train a trojaned network to fool the Neural Cleanse detector (Wang et al., 2019), but their approach is not applicable to other detection methods. Goldwasser et al. (2022) examine the problem from a cryptographic perspective and find that for one-layer networks it is possible to construct trojans that are computationally infeasible to detect. Most similar to our work is that of Sahabandu et al. (2022), who train trojans and a meta-network detector in a min-max alternating fashion to be hard to distinguish from clean networks.

We depart from prior work by developing evasive trojans that generalize to a wide range of detectors without specifically optimizing to fool them. Instead, we give detectors the upper hand in our evaluations by allowing access to training datasets of networks containing our evasive trojans. Additionally, we are the first to systematically measure reverse-engineering on a large scale, which allows us to make the surprising discovery that evasive trojans are also much harder to reverse-engineer.

## 3 BACKGROUND

**Neural Trojans.** A neural trojan is described by a trigger that can be applied to the inputs of a victim network and a hidden behavior that the trigger should activate in the victim network. For simplicity, we focus on classification networks and all-to-one attacks, where inserting a trigger reliably causes the victim network to output a fixed class. Let $C$ be the number of classes, and let $f \colon \mathcal{X} \to \mathbb{R}^C$ be a victim network that maps inputs $x \in \mathcal{X}$ to their posterior prediction. An attack specification is a tuple $(q, h, c)$, where $q \in \mathcal{Q}$ is a trojan trigger, $h \colon \mathcal{X} \times \mathcal{Q} \to \mathcal{X}$ is a function that inserts triggers into inputs, and $c \in \{1, \dots, C\}$ is the target label of the attack. We also define distributions $P_X$ and $P_Q$ over $\mathcal{X}$ and $\mathcal{Q}$ to model the data distribution and the distribution of

triggers being considered by the adversary. The associated random variables are $X$ and $Q$.

A trojan is successfully inserted if the attack success rate (ASR) is high, where ASR is defined as $\mathbb{P}(\text{argmax}_{c'} f(h(X, q))_{c'} = c)$, the probability of a triggered input being classified as the target label. Other desirable properties of an attack include not affecting accuracy on clean inputs and having high specificity, where specificity refers to the ability of alternate triggers $q' \in \mathcal{Q} \setminus \{q\}$ to activate the hidden behavior. If a trojan has low specificity and the defender has some knowledge of $\mathcal{Q}$, then the trojan can be readily detected by sampling triggers and analyzing their effect on $f$. Prior works consider a weaker notion of specificity (Pang et al., 2022; Zhang et al., 2021; Ren Pang, 2019), where a trojan has high specificity if it does not impact accuracy on clean examples. We extend this to include examples with unintended triggers.

**Threat Model.** We model trojan detection as an interaction between an attacker and defender. The goal of the attacker is to insert a trojan into a victim network without being detected, and the goal of the defender is to detect whether the network contains a trojan. The attacker randomly samples their trigger and target label, and they may use various methods for inserting the trojan depending on their degree of access to the victim network. In data poisoning attacks, the attacker can poison a small fraction of the training set, while in training-time attacks they have full control over the dataset and training function.

Importantly, we assume that the defender has access to a dataset of clean and trojaned networks, where the trojans are inserted using the same method as the attacker but with random triggers $q \sim Q$ and target labels $c \in \{1, \ldots, C\}$. In other words, the defender knows what the attacker's distribution of trojans looks like, but they do not know the specific trigger or target label used by the attacker. We make this assumption because we are interested in studying whether trojans in deep neural networks can be made difficult to detect not just for one detector, but for all detectors.

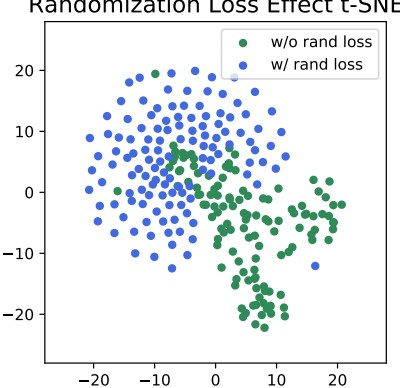

Figure 3: Top: Our distribution matching loss successfully maintains a tight coupling between evasive trojans $\theta_f$ and clean initializations $\theta_g$ and can thus be interpreted as minimizing the 1-Wasserstein distance. Bottom: Omitting the randomization loss leads to emergent coordination in the differences between summary statistics $\theta'_f - \theta'_g$, which cluster in one direction. With the randomization loss, the coordination disappears.

# 4 EVASIVE TROJANS

We develop a general method for inserting hard-to-detect trojans into deep neural networks. This method is detector-agnostic and can be applied to a wide variety of underlying trojan attacks, referred to as "standard trojans". Starting with a standard trojan attack defined by an attack specification $(q, h, c)$, the form of our loss for training evasive trojans is $\mathcal{L}_{\text{task}} + \mathcal{L}_{\text{trojan}} + \mathcal{L}_{\text{evasion}}$, where $\mathcal{L}_{\text{task}}$ is the task loss that increases accuracy on clean examples and $\mathcal{L}_{\text{trojan}}$ is the trojan loss that increases the attack success rate, and $\mathcal{L}_{\text{evasion}}$ is the evasion loss, which is designed to make trojans hard to detect. As with standard trojans, the task loss and trojan loss are implemented via cross-entropy on clean examples and examples with triggers inserted. The main modification for evasive trojans is the evasion loss, which we describe below.

## 4.1 EVASION LOSS

We identify three high-level components for an evasion loss: distribution matching, specificity, and randomization. The core of our approach is our distribution-matching loss, which enforces similarity between the distribution of clean networks and trojaned networks. The specificity and randomization

losses augment this central loss by addressing two practical challenges with designing hard-to-detect trojans for deep neural networks.

**Distribution Matching.** A natural approach to making trojans hard to detect is to ensure that across triggers, target labels, and other sources of randomness, the distribution of the resulting trojaned networks is similar to the distribution of clean networks. One way of accomplishing this would be to use a GAN loss. However, for this to be effective one would need to train a generator of networks or a diverse population of trojaned networks in parallel, which is computationally prohibitive for the attacker.

|  |  | ASR | Accuracy |
|---|---|---|---|
| Clean Networks | MNIST |  | 99.3 |
|  | CIFAR-10 |  | 94.0 |
|  | CIFAR-100 |  | 74.6 |
|  | GTSRB |  | 84.7 |
| Standard Trojans | MNIST | 100.0 | 99.3 |
|  | CIFAR-10 | 99.8 | 93.9 |
|  | CIFAR-100 | 98.7 | 74.5 |
|  | GTSRB | 97.2 | 84.5 |
| Evasive Trojans | MNIST | 99.4 | 99.2 |
|  | CIFAR-10 | 97.9 | 93.9 |
|  | CIFAR-100 | 98.7 | 74.6 |
|  | GTSRB | 97.1 | 84.0 |

Table 1: Attack success rate (ASR) and task accuracy. All values are percentages. ASR is averaged across patch and blended attacks. Task accuracy is similar across all networks for each dataset. Both standard and evasive trojans achieve high ASR.

Our key insight is that instead of using a loss based on GANs, we can use a loss based on the primal form of the 1-Wasserstein distance. Let $F$ and $G$ be random variables corresponding to distributions of trojaned and clean networks, respectively. The 1-Wasserstein distance is defined as $W_1(F, G) = \inf_\Gamma \mathbb{E}_{(F,G)\sim\Gamma} [d(F, G)]$, where $\Gamma$ is a coupling between $F$ and $G$—a joint distribution with marginals equal to $P_F$ and $P_G$—and $d$ is a distance metric. In general, finding the infimum over all couplings is challenging. However, we can approximate the infimum by arbitrarily fixing a coupling and maintaining it throughout training. If the distances remain small throughout training, it will remain a faithful approximation.

In practice, this corresponds to first training a clean network $g \sim G$, then initializing the trojan network $f$ from the parameters of $g$ and maintaining a small distance between the two networks according to a distance metric $d$. The selection of $d$ is an important hyperparameter. In preliminary experiments, we found that a simple combination of $\ell_2$ distance in parameter space and $\ell_1$ distance in the final unnormalized logits on clean examples was sufficient to see a notable increase in evasiveness, so this is what we use throughout the paper. Concretely, our distribution matching loss is

$$\mathcal{L}_{\text{dist}} = \lambda_1 \|\theta_f - \theta_g\|_2 + \lambda_2 \mathbb{E}_X \left[\|f'(X) - g'(X)\|_1\right],$$

where $\theta_f, \theta_g$ are the parameters of $f$ and $g$, the functions $f', g'$ output unnormalized logits, and $\lambda_1, \lambda_2$ are weights for adjusting the strength of the two distances.

**Specificity.** Under our threat model, the defender has access to a training dataset of clean and trojaned models. In some cases, they may also have access to the triggers accompanying the trojaned models in their training set. If the attacker's trojans have low specificity and respond to many unintended triggers, they can become trivial to detect by simply inserting the available triggers into clean inputs and analyzing their effect on a given network $f$.

In experiments, we find that low specificity is a significant problem for trojan attacks on deep neural networks, possibly because high-specificity trojans require more complex feature detectors to filter out unintended trigger patterns. This motivates us to add a loss encouraging high specificity. Let $q \in \mathcal{Q}$ be the trigger used for a trojan. The general approach for a specificity loss involves inserting incorrect triggers $q' \in \mathcal{Q} \setminus \{q\}$ into training examples and enforcing normal behavior on those "negative examples". Prior works with specificity losses have used cross-entropy to the clean label on negative examples (Nguyen & Tran, 2021). However, we find that optimizing this loss introduces additional discrepancies, since clean networks do not behave normally on examples with triggers inserted. We find that a more effective loss is to match posteriors between the trojaned network $f$ and its clean initialization $g$ on negative examples. Concretely, our specificity loss is

$$\mathcal{L}_{\text{specificity}} = \mathbb{E}_{X,Q} \left[\text{cross-entropy}(f(h(X, Q)), g(h(X, Q)))\right],$$

where $h$ is the trigger insertion function. This can be interpreted as a component of the distribution matching loss, but it serves a distinct purpose of increasing the specificity of the trojan.

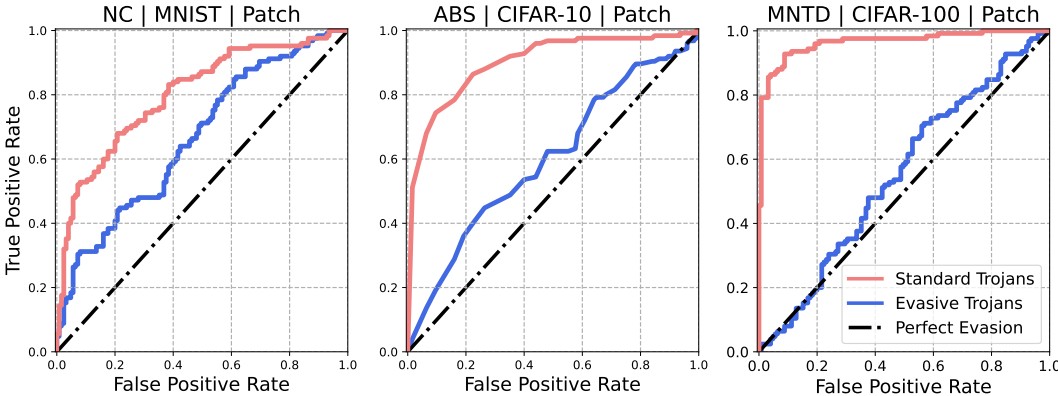

Figure 4: ROC curves for standard trojans and our evasive trojans across a variety of detectors and datasets. In some cases, evasive trojans reduce detection performance to near-chance levels.

**Randomization.** Empirically, we find that the distribution matching loss greatly increases evasiveness against many detectors. However, a simple logistic regression on summary features of the network parameters is able to detect trojans using $\mathcal{L}_{\text{dist}}$ surprisingly well—even better than state-of-the-art detectors. This suggests that the distribution matching loss induces emergent coordination in the parameter space, such that independently trained evasive trojans acquire similarities. To mitigate this problem, we propose an additional randomization loss $\mathcal{L}_{\text{rand}} = 1 - \text{cosine}(\theta'_f - \theta'_g, r)$, where $\theta'_f, \theta'_g$ are the summary statistics for $f$ and $g$, and $r$ is a random direction fixed at the start of training. This loss encourages the difference between $\theta_f$ and $\theta_g$ to not only be small, but to also point in a random direction. Note that enforcing this loss directly in the feature space is too strict, so we use summary statistics instead. See Appendix A for more details. In Figure 3, we visualize the effect of the distribution matching and randomization losses.

## 5 EXPERIMENTS

To evaluate our approach for creating evasive trojans, we ask whether trojans with a fixed distribution of triggers can be made harder to detect for a variety of detectors, ranging from simple baselines to state-of-the-art detection algorithms. We generate collections of clean networks, networks with standard trojans, and networks with evasive trojans. We train these networks on MNIST, CIFAR-10, CIFAR-100, and GTSRB (downsampled to 32x32). For each dataset, we generate 250 clean networks and 250 networks for each trojan attack, split evenly into training and test sets. The training sets of networks are used for developing trojan detectors. These detectors are then evaluated on the test networks. In total, we train over 6,000 neural networks to evaluate our method.

**Trojan Attacks.** We consider two foundational trojan attacks from the literature: the *Patch* attack (Gu et al., 2017) and *Blended* attack (Chen et al., 2017). The patch attack uses small patches with binary patterns as triggers, and the trigger insertion function stamps them onto the input image. Each patch trigger consists of a pattern and a mask indicating where the pattern should go in an input image. We sample patch triggers with masks of varying shapes and locations, but the mask for each trojan is static once the location is initially decided. The blended attack is a more subtle attack, using image-wide random noise as the trigger, which is blended into the input image at a low magnitude. Both attacks are implemented by poisoning a small percentage of the training data. We poison 1% of the training set and refer to these attacks as *Standard Trojans*.

We train networks with *Evasive Trojans* using the procedure described in Section 4. First, we train a new dataset of clean networks for initializing the evasive trojans. These are independent from the clean networks used for training and evaluating detectors and are only used for initializing and training evasive trojans. Next, we train the evasive trojans using the same patch and blended triggers that the standard trojans use. The trigger distribution is fixed, so the method for inserting trojans is the main independent variable. In Table 1, we show that the attack success rate is similar for standard trojans and our evasive trojans.

| | | | Acc | Spec | NC | ABS | Param | MNTD | Max | Avg |
|---|---|---|---|---|---|---|---|---|---|---|
| **Standard Trojans** | MNIST | Patch | 53.0 | 64.8 | 80.2 | 51.8 | 55.4 | 69.3 | 80.2 | 61.1 |
| | | Blended | 53.0 | 100.0 | 100.0 | 83.1 | 72.6 | 91.7 | 100.0 | 81.7 |
| | CIFAR-10 | Patch | 55.8 | 100.0 | 80.0 | 90.0 | 57.6 | 99.4 | 100.0 | 76.7 |
| | | Blended | 63.6 | 100.0 | 100.0 | 82.0 | 83.0 | 100.0 | 100.0 | 85.7 |
| | CIFAR-100 | Patch | 57.9 | 99.9 | 84.9 | 70.8 | 61.8 | 96.5 | 99.9 | 75.0 |
| | | Blended | 61.3 | 100.0 | 100.0 | 72.0 | 85.2 | 99.8 | 100.0 | 83.7 |
| | GTSRB | Patch | 50.3 | 71.0 | 64.0 | 56.2 | 48.5 | 63.3 | 71.0 | 58.0 |
| | | Blended | 51.4 | 78.5 | 100.0 | 60.9 | 99.9 | 96.8 | 100.0 | 78.2 |
| | Average | | 55.8 | 89.3 | 88.6 | 70.8 | **70.5** | 89.6 | 93.9 | 75.0 |
| **Evasive Trojans** | MNIST | Patch | 55.6 | 54.3 | 66.5 | 51.1 | 70.6 | 53.0 | 70.6 | 58.5 |
| | | Blended | 60.2 | 67.8 | 99.2 | 54.9 | 84.8 | 67.2 | 99.2 | 72.4 |
| | CIFAR-10 | Patch | 61.3 | 67.4 | 58.1 | 60.0 | 52.2 | 62.3 | 67.4 | 60.2 |
| | | Blended | 53.5 | 67.2 | 100.0 | 84.0 | 79.7 | 93.3 | 100.0 | 79.6 |
| | CIFAR-100 | Patch | 54.9 | 50.4 | 61.1 | 50.7 | 61.6 | 55.0 | 61.6 | 55.6 |
| | | Blended | 54.4 | 65.1 | 100.0 | 64.6 | 91.7 | 76.1 | 100.0 | 75.3 |
| | GTSRB | Patch | 50.8 | 73.7 | 56.6 | 54.8 | 77.1 | 48.7 | 77.1 | 60.3 |
| | | Blended | 55.0 | 72.3 | 100.0 | 81.3 | 85.5 | 62.0 | 100.0 | 76.0 |
| | Average | | **55.7** | **64.8** | **80.2** | **62.7** | 75.4 | **64.7** | **84.5** | **67.2** |

Table 2: Detection results. Our evasive trojans are harder to detect across a wide range of detectors, datasets, and attack specifications. All values are percent AUROC, and lower is better for the attacker. For each detector, we bold the better value in the "Average" row.

**Network Architectures and Hyperparameters.** For CIFAR-10 and CIFAR-100, we use 40-2 Wide ResNets (Zagoruyko & Komodakis, 2016) with a dropout rate of 0.3 (Srivastava et al., 2014). For GTSRB, we use the SimpleViT Vision Transformer (Beyer et al., 2022) as implemented by lucidrains. For MNIST, we use a simple 5-layer convnet with batch norm.

We train all CIFAR-10, CIFAR-100, and GTSRB networks for 50 epochs with a batch size of 128. We train all MNIST networks for 10 epochs with a batch size of 256 except for evasive trojans, which we found benefited from 20 epochs of training after initializing from clean networks.

We train all CIFAR-10 and CIFAR-100 networks using SGD with learning rate 0.1, weight decay of $5 \times 10^{-4}$, and Nesterov momentum of 0.9. We train all MNIST and GTSRB networks using Adam with a weight decay of $1 \times 10^{-5}$ and other hyperparameters at default settings. All training hyperparameters were chosen early in preliminary experiments and received minimal tuning.

**Detectors.** To measure the evasiveness of different trojans, we use a variety of detectors. The accuracy-based detector (*Acc*) simply uses the clean accuracy of a network as a score for detection. The specificity-based detector (*Spec*) computes the entropy of average posteriors on inputs with random triggers inserted. If a trojan has low specificity, these random triggers will activate it and reveal its presence. Neural Cleanse (*NC*) iterates over possible target labels for an attack and directly searches for candidate triggers using gradient-based optimization (Wang et al., 2019). *ABS* monitors the behavior of neurons while perturbing the input to identify candidate compromised neurons. These candidate neurons are used to generate a trojan trigger similarly to Neural Cleanse, and the maximum attack success rate across candidates is used as a detection score (Liu et al., 2019). *MNTD* trains a set of queries and a shallow classifier on the training set of clean and trojaned models (Xu et al., 2021). The *Param* detector is a logistic regressor with a single linear layer that takes summary statistics of the raw network parameters as its input. Finally, the *Max* and *Avg* baselines are the maximum and average AUROC obtained by the six detectors on a given set of trojaned networks.

## 5.1 DETECTION

To measure the effectiveness of detectors, we use area under the ROC curve (AUROC) on test sets of clean and trojaned networks. AUROC is a threshold-independent metric that can be interpreted as the probability that a positive example has a higher detection score than a negative example (Fawcett, 2006), so 50% corresponds to random detection performance. For hand-crafted detectors that do not leverage the training set, the AUROC can sometimes be below 50%. We find that this happens to a

|  |  |  | NC | ABS | Param | MNTD | Max | Avg |
|---|---|---|---|---|---|---|---|---|
| Standard Trojans | MNIST | Patch | 60.8 | 16.8 | 8.0 | 40.0 | 60.8 | 31.4 |
|  |  | Blended | 100.0 | 41.6 | 8.8 | 98.4 | 100.0 | 62.2 |
|  | CIFAR-10 | Patch | 52.0 | 94.4 | 11.2 | 99.2 | 99.2 | 64.2 |
|  |  | Blended | 98.4 | 84.8 | 11.2 | 100.0 | 100.0 | 73.6 |
|  | CIFAR-100 | Patch | 38.4 | 70.4 | 0.0 | 28.8 | 70.4 | 34.4 |
|  |  | Blended | 100.0 | 48.0 | 0.0 | 14.4 | 100.0 | 40.6 |
|  | GTSRB | Patch | 35.2 | 19.2 | 3.2 | 9.6 | 35.2 | 16.8 |
|  |  | Blended | 100.0 | 32.0 | 3.2 | 46.4 | 100.0 | 45.4 |
|  | Average |  | 73.1 | 50.9 | 5.7 | 54.6 | 83.2 | 46.1 |
| Evasive Trojans | MNIST | Patch | 28.8 | 13.6 | 8.0 | 17.6 | 28.8 | 19.4 |
|  |  | Blended | 92.0 | 28.0 | 9.6 | 68.8 | 92.0 | 58.1 |
|  | CIFAR-10 | Patch | 9.6 | 40.0 | 11.2 | 12.8 | 40.0 | 22.7 |
|  |  | Blended | 7.2 | 80.8 | 9.6 | 88.8 | 88.8 | 55.0 |
|  | CIFAR-100 | Patch | 1.6 | 2.4 | 0.0 | 0.8 | 2.4 | 1.4 |
|  |  | Blended | 2.4 | 34.4 | 1.6 | 8.8 | 34.4 | 16.3 |
|  | GTSRB | Patch | 1.6 | 20.0 | 1.6 | 3.2 | 20.0 | 9.3 |
|  |  | Blended | 3.2 | 76.0 | 1.6 | 19.2 | 76.0 | 35.2 |
|  | Average |  | **18.2** | **36.9** | **5.6** | **27.3** | **47.8** | **27.2** |

Table 3: Target label prediction results. Although we do not specifically design our evasive trojans to be hard to reverse-engineer, we find that predicting their target labels is much harder. All values are percent accuracy, and lower is better for the attacker. These are unexpected and concerning results that highlight the need for more robust trojan detection and reverse-engineering methods.

small degree in some experiments. In these cases, we negate the detection score before computing AUROC on the test set.

**Main Results.** Detection results are in Table 2, and sample ROC curves are in Figure 4. We train standard and evasive trojans in eight settings and evaluate them on six detectors. Average AUROC across all eight settings is lower for evasive trojans in five out of the six detectors. In some cases, evasiveness substantially improves. For example, average AUROC for the MNTD detector drops by 25%. When looking at the most effective detector in each setting, evasiveness also improves on average, with a 9.4 percent drop in AUROC. This shows that our evasive trojans are harder to detect not just for a specific detector, but for a diverse range of detectors that use different mechanisms.

Surprisingly, the blended attacks are detected very easily by Neural Cleanse, and our evasion loss is unable to reduce the efficacy of Neural Cleanse in these settings. This is unexpected, because Neural Cleanse is designed specifically to detect patch attacks. However, our evasion loss does make blended attacks harder to detect for other methods, including MNTD and in some settings ABS. As shown in Figure 2, although blended attacks tend to be easier to detect than patch attacks, evasive trojans reduce the efficacy of the average detector across all four datasets. Additional results and experiment details are in Appendix B.

## 5.2 REVERSE-ENGINEERING

Once a trojan has been detected, one might want to know what the intended functionality of the trojan is or what causes it to activate. Reverse-engineering trojans is a nascent field with relatively little prior work. However, since evasive trojans make detection more challenging, a natural question to ask is whether they also make reverse-engineering harder. We operationalize these reverse-engineering tasks as predicting the target label of a trojan attack and predicting the segmentation mask of patch attacks. Since recovering trigger patterns is nontrivial (Guo et al., 2019), we focus on reverse-engineering the trigger mask.

**Target Label Prediction.** We use accuracy as a metric for predicting target labels. Neural Cleanse and ABS predict target labels as part of their detection pipeline, so no modification is needed. For the MNTD and Param detectors, we replace their output layer and train them as classifiers with multiclass cross-entropy. Results are in Table 3. Surprisingly, we find that evasive trojans are not only harder to detect, but they also make predicting the target label considerably harder. For each of

|  |  | NC | ABS | Param | MNTD | Max | Avg |
|---|---|---|---|---|---|---|---|
| Standard Trojans | MNIST | 4.9 | 4.5 | 4.6 | 3.8 | 4.9 | 4.4 |
|  | CIFAR-10 | 6.0 | 4.6 | 5.5 | 7.6 | 7.6 | 5.9 |
|  | CIFAR-100 | 6.4 | 5.0 | 7.6 | 7.1 | 7.6 | 6.5 |
|  | GTSRB | 5.5 | 6.5 | 7.2 | 5.6 | 7.2 | 6.2 |
|  | Average | **5.7** | **5.2** | 6.2 | 6.0 | 6.8 | 5.8 |
| Evasive Trojans | MNIST | 5.7 | 5.3 | 5.9 | 5.2 | 5.9 | 5.5 |
|  | CIFAR-10 | 5.7 | 4.3 | 4.1 | 4.8 | 5.7 | 4.7 |
|  | CIFAR-100 | 5.9 | 5.6 | 4.8 | 5.2 | 5.9 | 5.4 |
|  | GTSRB | 5.6 | 6.0 | 7.2 | 4.0 | 7.2 | 5.7 |
|  | Average | 5.7 | 5.3 | **5.5** | **4.8** | 6.2 | **5.3** |

Table 4: Trigger synthesis results. All values are percent IoU, and lower is better for the attacker. Although IoU is low across the board, evasive trojans further reduce IoU for the most effective methods. This demonstrates the need to develop stronger and more robust trigger synthesis methods.

the four classifiers, accuracy on evasive trojans is lower. Notably, the average accuracies for Neural Cleanse, ABS, and MNTD drop by $54.9$, $14$ and $27.3$ percentage points, respectively. The accuracy of the best classifier in each setting drops by $35.4\%$ on average.

Accuracy on evasive trojans drops to chance levels in several settings. For example, on CIFAR-10 standard trojans, MNTD reaches $99.2\%$ accuracy, but for evasive trojans it drops to $11.2\%$ accuracy (random chance would be $10\%$). As with detection, the classifiers are more effective on blended attacks, but Neural Cleanse is also reduced from near-perfect prediction to random chance on blended attacks. Our evasion loss was only intended to make trojans harder to detect, and there is no *a priori* reason for it to make target labels hard to predict. Consequently, this is a very unexpected and concerning result for defense methods.

**Trigger Synthesis.** We use mean intersection over union (IoU) across trojaned networks as a metric for predicting trigger masks. Neural Cleanse and ABS generate candidate trigger masks as part of their detection pipeline, so no modification is needed. For MNTD and Param, we replace the output layer with a $4$-dimensional output that regresses to the top-left and bottom-right coordinates of trigger masks in the training set. If a predicted bounding box is invalid, the predicted mask defaults to the entire image. In all trigger synthesis experiments, only patch attacks are used. The trigger masks have varying shapes and locations, but they are fixed upon sampling for a given trojan. Thus, the task is well-defined and is a standard binary segmentation task.

Results are in Table 4. In general, performance is quite poor across the trigger synthesis methods, with IoU never reaching above $8\%$. Additionally, average IoU is very close for standard trojans and evasive trojans on Neural Cleanse and ABS. However, average IoU for Param and MNTD is decreased by evasive trojans. For MNTD, IoU drops from $6\%$ to $4.8\%$, which is a $20\%$ relative reduction. The IoU of the most effective trigger synthesis method drops from $6.8\%$ to $6.2\%$ on average. These results indicate that trigger synthesis is somewhat more difficult on evasive trojans. However, IoU values are close to the floor in all cases, which demonstrates a need for more research on this important aspect of reverse-engineering trojans.

## 6    CONCLUSION

We introduced a method for inserting evasive trojans in deep neural networks. Unlike standard trojan attacks, our evasive trojans are specifically designed to be hard to detect. To evaluate our method, we trained standard and evasive trojans on a large scale, creating training and test sets containing over $6,000$ neural networks. These networks were used to train and evaluate a wide variety of trojan detectors, including state-of-the-art detection algorithms and simple yet effective baselines. We found that our evasive trojans are much harder to detect across a wide range of evaluation settings, in some cases reducing detection performance to chance levels. Surprisingly, we found that our evasive trojans also make reverse-engineering the target label and trigger of a trojan attack substantially harder. We hope these results demonstrate the need for further research into robust mechanisms for monitoring and detecting hidden functionality in deep neural networks.

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

## A  EVASIVE TROJANS DETAILS

**Randomization Loss.**  The randomization loss minimizes the cosine distance between the network parameters and a random direction. However, using the randomization loss in the raw parameter space is far too restrictive and effectively amounts to adding noise to the parameters. Thus, we use a space of differentiable summary features of the parameters instead. We concatenate the mean and standard deviation of each parameter vector obtained via the PyTorch parameters enumerator, which forms a single vector summarizing the network parameters.

For MNIST networks, we found that even this loss was too restrictive and hard to satisfy, perhaps due to the smaller size of the networks. Thus, we use an alternate randomization loss for MNIST networks. Let $\theta'_f$ be summary statistics of the trojaned model parameters, let $\theta'_g$ be summary statistics of the clean initialization, and let $r$ be a binary random vector. The alternate randomization loss is

$$\mathcal{L}_{\text{rand}} = \| \min(\theta'_f - \theta'_g, \mathbf{0}) \odot r \|_1 + \| \min(\theta'_g - \theta'_f, \mathbf{0}) \odot (\mathbf{1} - r) \|_1,$$

where the min function outputs the elementwise minimum. This loss allows each summary statistic in the trojaned network to either be greater or smaller than the corresponding summary statistic in the clean initialization, where the direction of allowed change is selected by the binary random vector. This can be considered a softer, axis-aligned version of the original randomization loss.

**Hyperparameters.**  Our evasive trojans have several hyperparameters. The two components of the distribution matching loss and the randomization loss each receive a separate weight. Let these weights be denoted by $\lambda_1$, $\lambda_2$, and $\lambda_3$, respectively. We use different loss weights for different experiments, and the weights are coarsely tuned primarily to ensure that ASR is high enough. Empirically, we find that increasing the weights for these losses smoothly decreases ASR and eventually the task accuracy. This is because the evasion loss is in direct conflict with the trojan loss, so there is a balancing act between satisfying the evasion loss and maintaining high ASR. Concretely, we use the following settings of the loss weights:

- For blended attacks on CIFAR-10, $\lambda_1 = 0.02$, $\lambda_2 = 0.02$, $\lambda_3 = 0.00005$
- For blended attacks on CIFAR-100, $\lambda_1 = 0.01$, $\lambda_2 = 0.02$, $\lambda_3 = 0.0001$
- For blended attacks on GTSRB, $\lambda_1 = 0.005$, $\lambda_2 = 0.01$, $\lambda_3 = 0.00001$
- For patch attacks on GTSRB, $\lambda_1 = 0.02$, $\lambda_2 = 0.02$, $\lambda_3 = 0.0001$
- For patch attacks on CIFAR-10, $\lambda_1 = 0.02$, $\lambda_2 = 0.02$, $\lambda_3 = 0.00005$
- For all other experimental settings, $\lambda_1 = 0.1$, $\lambda_2 = 0.05$, $\lambda_3 = 0.0001$

We tune these weights by training small numbers of individual networks and monitoring the evasion loss components and ASR. Importantly, we do not tune them specifically to obtain better results in the main experiments except in preliminary experiments meant to identify appropriate ranges for the losses, which we performed in a small number of settings. In a few experimental settings, we observed that there was a long tail of networks with low ASR. We hypothesized that this was due to the randomization loss picking a challenging direction. Thus, we retrained all networks below a cutoff ASR using new random directions, which solved the problem. In general, we find that our evasion loss is fairly robust to selections of loss weights and easy to use once the appropriate ranges for the weights are identified. The specificity loss is implemented by inserting incorrect triggers into 16 examples for blended attacks and 10 examples for patch attacks. These numbers were selected early during preliminary experiments.

|  |  |  | ASR | Accuracy |
|---|---|---|---|---|
| Clean Networks | MNIST | | | 99.3 |
| | CIFAR-10 | | | 94.0 |
| | CIFAR-100 | | | 74.6 |
| | GTSRB | | | 84.7 |
| | Average | | | 88.1 |
| Standard Trojans | MNIST | Patch | 100.0 | 99.3 |
| | | Blended | 100.0 | 99.3 |
| | CIFAR-10 | Patch | 100.0 | 93.9 |
| | | Blended | 99.5 | 93.9 |
| | CIFAR-100 | Patch | 99.8 | 74.5 |
| | | Blended | 97.5 | 74.5 |
| | GTSRB | Patch | 99.8 | 85.5 |
| | | Blended | 94.6 | 83.5 |
| | Average | | 98.9 | 88.0 |
| Evasive Trojans | MNIST | Patch | 99.5 | 99.3 |
| | | Blended | 99.2 | 99.2 |
| | CIFAR-10 | Patch | 100.0 | 93.9 |
| | | Blended | 95.8 | 94.0 |
| | CIFAR-100 | Patch | 99.9 | 74.6 |
| | | Blended | 97.4 | 74.7 |
| | GTSRB | Patch | 96.4 | 84.4 |
| | | Blended | 97.8 | 83.5 |
| | Average | | 98.3 | 87.9 |

Table 5: Attack success rate (ASR) and task accuracy in all experimental settings. Each value is averaged across 125 neural networks in the validation set for the indicated experimental setting. All values are percentages.

**Other Details.** In preliminary experiments, we found that several implementation details were important for increasing the evasiveness of our trojans. Namely, we train all evasive trojans without dropout. Clean initializations are trained with dropout, but during the second stage of training we turn dropout off. This is because dropout introduces uncorrelated randomness in the activations of the trojaned network and its clean initialization, which makes satisfying the logit matching component of $\mathcal{L}_{\text{dist}}$ challenging. For similar reasons, we also switch batch norm layers in clean initialization networks to eval mode throughout the second stage of training evasive trojans.

To improve performance on blended attacks, we found that it was important to process the inputs for the clean, trojan, and specificity losses together in a single forward pass. This is because networks that use batch norm are able to "cheat" by aggregating information across the batch. Empirically, this issue arose most prominently with blended attacks. Concatenating the inputs together fixes the problem.

## B  ADDITIONAL RESULTS

**Additional Details on Detectors.**

• The accuracy-based detector (*Acc*) simply uses the clean accuracy of a network as a score for detection. If a trojan insertion method consistently decreases clean accuracy, it can become trivial to detect, so this is an important baseline detector.

• The specificity-based detector (*Spec*) assumes that the defender has access to a small set of $k$ triggers sampled from the same distribution of triggers that are used by the trojaned networks in question. This detector inserts each of the $k$ triggers into images from the validation set and computes the entropy of the average posterior. The $k$ entropy values are then averaged, the negative of

|  |  |  | Acc | Spec | NC | ABS | Param | MNTD | Max | Avg |
|---|---|---|---|---|---|---|---|---|---|---|
| Without $\mathcal{L}_{\text{rand}}$ | MNIST | Patch | 56.5 | 53.4 | 63.1 | 53.6 | 67.7 | 60.9 | 67.7 | 59.2 |
|  |  | Blended | 58.4 | 54.1 | 97.3 | 61.4 | 93.6 | 74.4 | 97.3 | 73.2 |
|  | CIFAR-10 | Patch | 72.8 | 71.1 | 54.7 | 61.3 | 85.7 | 88.6 | 88.6 | 72.4 |
|  |  | Blended | 57.4 | 66.7 | 100.0 | 90.8 | 100.0 | 91.3 | 100.0 | 84.4 |
|  | CIFAR-100 | Patch | 74.1 | 98.8 | 55.7 | 54.1 | 100.0 | 74.9 | 100.0 | 76.3 |
|  |  | Blended | 50.0 | 72.2 | 100.0 | 74.1 | 100.0 | 94.5 | 100.0 | 81.8 |
|  | GTSRB | Patch | 51.4 | 62.6 | 54.5 | 53.0 | 78.2 | 49.5 | 78.2 | 58.2 |
|  |  | Blended | 52.2 | 55.4 | 100.0 | 84.5 | 93.5 | 74.8 | 100.0 | 76.7 |
|  | Average |  | 59.1 | 66.8 | 78.2 | 66.6 | 89.8 | 76.1 | 91.5 | 72.8 |
| With $\mathcal{L}_{\text{rand}}$ | MNIST | Patch | 55.6 | 54.3 | 66.5 | 51.1 | 70.6 | 53.0 | 70.6 | 58.5 |
|  |  | Blended | 60.2 | 67.8 | 99.2 | 54.9 | 84.8 | 67.2 | 99.2 | 72.4 |
|  | CIFAR-10 | Patch | 61.3 | 67.4 | 58.1 | 60.0 | 52.2 | 62.3 | 67.4 | 60.2 |
|  |  | Blended | 53.5 | 67.2 | 100.0 | 84.0 | 79.7 | 93.3 | 100.0 | 79.6 |
|  | CIFAR-100 | Patch | 54.9 | 50.4 | 61.1 | 50.7 | 61.6 | 55.0 | 61.6 | 55.6 |
|  |  | Blended | 54.4 | 65.1 | 100.0 | 64.6 | 91.7 | 76.1 | 100.0 | 75.3 |
|  | GTSRB | Patch | 50.8 | 73.7 | 56.6 | 54.8 | 77.1 | 48.7 | 77.1 | 60.3 |
|  |  | Blended | 55.0 | 72.3 | 100.0 | 81.3 | 85.5 | 62.0 | 100.0 | 76.0 |
|  | Average |  | 55.7 | 64.8 | 80.2 | 62.7 | 75.4 | 64.7 | 84.5 | 67.2 |

Table 6: Randomization loss ablation. Without the randomization loss, the Param detector is especially strong, leading to a high maximum AUROC across all detectors. Adding the randomization loss greatly reduces AUROC for MNTD and Param detectors. For the other detectors, average AUROC remains nearly unchanged. All values are percent AUROC, and lower is better for the attacker.

which is used as the detection score. For trojans with low specificity, the entropy of the average posterior for triggered inputs will be lower than for clean networks, which enables detection.

- Neural Cleanse (*NC*) iterates over possible target labels for an attack and directly searches for candidate triggers using gradient-based optimization (Wang et al., 2019). We use a simplified version of Neural Cleanse that we found obtains stronger detection performance. Namely, in preliminary experiments we found that early stopping did not improve results, so we optimize for a fixed number of gradient steps. Additionally, the original Neural Cleanse method uses an anomaly index based off of the $\ell_1$ norms of the optimized trigger masks for detection, which enables selecting a principled threshold. However, we find that simply using the raw $\ell_1$ norms results in significantly better detection, so we switch to this simpler score for Neural Cleanse. This is enabled by our large-scale evaluations on datasets of clean and trojaned networks, which allows using threshold-independent metrics and any real-valued detection score.

- *ABS* monitors the behavior of neurons while perturbing the input to identify candidate compromised neurons. These candidate neurons are used to generate a trojan trigger similarly to Neural Cleanse, and the maximum attack success rate across candidates is used as a detection score (Liu et al., 2019).

- *MNTD* consists of a set of query inputs, which are passed through the network in question. The outputs on these queries are then concatenated and passed to a shallow classifier, which outputs a detection score. The queries and shallow classifier are optimized on the training set of clean and trojaned networks (Xu et al., 2021). MNTD is an example of a broad class of techniques called meta-networks: neural networks trained to interpret or monitor other neural networks.

- The *Param* detector is a logistic regressor with a single linear layer that takes summary statistics of the raw network parameters as its input. For summary statistics, we concatenate the min, max, mean, median, and standard deviation of each parameter vector into a single feature vector summarizing the raw parameters of the network.

## B.1 ABLATIONS.

Our evasive trojan training procedure has several distinct components. Here, we examine what happens when certain components are removed.

|  |  | NC | Param | MNTD |
|---|---|---|---|---|
| With $\mathcal{L}_{\text{penultimate}}$ | Patch | 58.8 | 100 | 60.5 |
|  | Blended | 91.6 | 100 | 70.9 |
| Without $\mathcal{L}_{\text{penultimate}}$ | Patch | 66.5 | 70.6 | 53.0 |
|  | Blended | 99.2 | 84.8 | 67.2 |

Table 7: Evaluation of using an $\ell_1$ distance on the penultimate features as an additional component of the distance metric. Compared to the original distance metric, this improves evasiveness against Neural Cleanse (lower AUROC) but reduces evasiveness against MNTD and Param (higher AUROC). All values are percent AUROC, and lower is better for the attacker.

**Randomization Loss.** We include the randomization loss to mitigate emergent coordination across independently trained evasive trojans. This coordination occurs when only using the distribution-matching and specificity losses, and it enables strong detection performance with a simple detector that performs a logistic regression on summary statistics of the parameters (Param).

In Table 6, we compare evasive trojans with and without the randomization loss. When the randomization loss is removed, the Param and MNTD detectors become much stronger, while average AUROC for the other detectors remains relatively unchanged. In several cases for trojans without the randomization loss, the Param detector obtains 100% AUROC. Consequently, including the randomization loss substantially reduces the AUROC of the best detector from an average of 91.5% to 84.5%. These results demonstrate that the randomization loss is an important component of our method for training evasive trojans.

**Specificity Loss.** We include the specificity loss to prevent the issue of low specificity, where unintended triggers can activate the trojan. If a trojan has low specificity, then a defender with knowledge of the distribution of triggers can easily detect the trojan by checking whether the known triggers cause unusual behavior. Our specificity-based detector (Spec) is based on this intuition. To validate the importance of the specificity loss, we retrain the CIFAR-10 blended evasive trojans without the specificity loss. The specificity detector obtains 100% AUROC on these networks compared to 67.2% AUROC when the specificity loss is used. This indicates that the specificity loss has the desired effect and is an important component of our method for training evasive trojans.

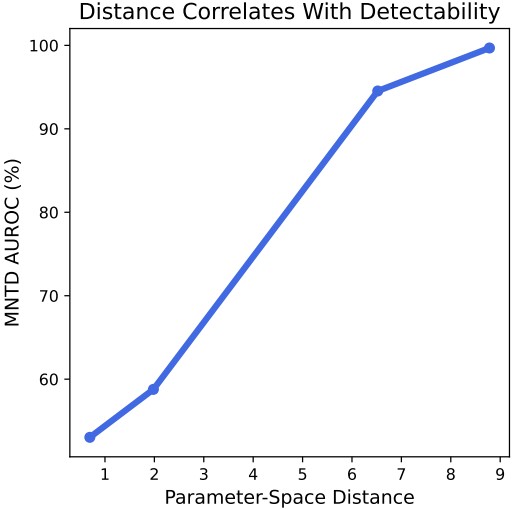

Figure 5: When training evasive trojans with different weights on the evasion loss, one can control the distance from the paired clean network in logit-space and parameter-space. These distances correlate with the detection performance of MNTD. This suggests that further reducing our current distance metric could lead to even greater evasiveness.

**Logit Matching Loss.** The logit matching loss is one of the two components of our distribution matching loss. To isolate the impact of this loss, we train retrain the CIFAR-10 patch evasive trojans without the logit matching loss. The MNTD detector obtains 70.8% AUROC on these networks compared to 62.3% with the logit matching loss and 99.4% for standard trojans. This shows that the logit matching loss is an important component of our evasive trojans, but it only accounts for part of the increased evasiveness.

**Different Distance Metrics.** Since the distance metric is an important component of our distribution-matching loss, an interesting question is what happens when the metric is changed. Here, we explore adding an $\ell_1$ distance on the penultimate features to the distance metric. Concretely, we add $\mathcal{L}_{\text{penultimate}} = \lambda_p \mathbb{E}_X \left[ \| f_p(X) - g_p(X) \|_1 \right]$, where $g_p$ and $f_p$ are functions that output

the penultimate features of the respective networks and $\lambda_p$ is a scalar loss weight. We set $\lambda_p$ to equal 0.1 and retrain the MNIST evasive trojans using the modified distance metric. As before, we train 500 models, split evenly into patch and blended triggers and divided into training and test sets. We evaluate these trojaned models against baseline detectors and show the results in Table 7. We find that that evasiveness against Neural Cleanse increases, but evasiveness against MNTD and Param decreases. This demonstrates that the distance metric has a large effect on evasiveness, and designing good distance metrics that improve evasiveness across many diverse detectors is nontrivial.

**Impact of Evasion Loss on Detector Performance.** Two natural questions are (1) whether our evasion loss actually reduces the distance in parameter and logit space as intended and (2) whether this correlates with improved evasiveness. To more precisely evaluate the impact of our evasion loss, we retrain our evasive trojans with patch triggers on MNIST using different weights on the evasion loss. For each training run, we multiply all components of the evasion loss by a fixed scalar ranging from 1 (original evasion loss) to 0 (no evasion loss, but still initializing from a clean network). In particular, the loss weights are 1, 0.01, 0.001, and 0. The corresponding distance values in parameter space are 0.7, 2.0, 6.5, and 8.8. In logit space, the distance values are 2.2, 2.5, 5.9, and 33.9, respectively. This shows that our evasion loss is optimized successfully. To see whether this translates into changes in detectability, we compute the percent AUROC for MNTD at each of these loss weights. In Figure 5, we show the results of this experiment by plotting distance in parameter-space on the x-axis and MNTD AUROC on the y-axis. There is a clear correlation: larger parameter distances result in higher detection performance. This suggests that evasiveness could be further improved by developing approaches that allow one to reduce our current distance metric even further.

### B.2 Additional Attack Comparisons

Here, we compare to other trojan attacks that are designed with evasiveness in mind. In each section, we clarify how these prior attacks differ from our own.

**Blind Backdoors Neural Cleanse Evasion Method.** Bagdasaryan & Shmatikov (2021) train trojans specifically to evade Neural Cleanse. Namely, a Neural Cleanse search process is carried out simultaneously with model training in a two-phase update approach. We implement this method and train a dataset of 500 MNIST models to evaluate its evasiveness. As with the standard and evasive trojans, we evenly split these models into patch and blended triggers and divide them into training and test sets.

With MNTD, the AUROC for patch and blended trojans is 72.8% and 98.7%, respectively. With Neural Cleanse, the AUROC for patch and blended trojans is 77.3% and 98.9%, respectively. With the Param detector, the AUROC for patch and blended trojans is 100.0% in both cases. Compared to the performance of Neural Cleanse on standard trojans, this is slightly better in both cases. This shows that their evasion method does work. However, MNTD and Param still have high performances on their trojans (in the case of Param, this reaches perfect detection performance). This shows that training trojans to be evasive for a specific detector may not generalize to all detectors. By contrast, our evasive trojans do generalize to reducing the detection performance of a broad range of detectors without specifically training against them.

**WaNet Warping Attack.** As we note in the related work, there have been numerous prior works exploring how to make trojan triggers more stealthy, which we distinguish from making trojans themselves more evasive. These methods are specifically designed to evade dataset-level and input-level detectors like Activation Clustering, Spectral Signatures, STRIP, and SentiNet. They do so by making inputs with triggers appear more similar to inputs without triggers (either in the input-space or intermediate features). However, these methods are not designed to evade model-level detectors like MNTD or ABS and are almost never evaluated on these detectors. An interesting question is whether the strong evasiveness of this class of trojans on dataset-level and input-level detectors transfers to evasiveness on model-level detectors. To investigate this, we train 500 trojaned CIFAR-10 models using the WaNet attack (Nguyen & Tran, 2021). This attack uses subtle spatial warping of the input as a trigger, which improves evasiveness against input-level detectors like STRIP.

We evaluate our baseline detectors against the trained WaNet models. The Neural Cleanse, MNTD, and Param detectors obtain AUROC scores of $99.5\%$, $100.0\%$, and $99.98\%$, respectively. Thus, they are very easy to detect. The result on Neural Cleanse runs counter to Neural Cleanse experiments in the WaNet paper. We are not certain what the cause for this discrepancy is, although there are several possible sources: (1) We use a custom PyTorch implementation of Neural Cleanse that uses a different detection score due to our evaluations being threshold-independent, (2) Many papers that propose attacks with whole-image triggers observe low detection performance with Neural Cleanse [cite, cite, cite]. However, our implementation of Neural Cleanse obtains very high AUROC on blended triggers. This is itself unexpected, but it could explain why our Neural Cleanse implementation also works for whole-image warping triggers. We tried out different hyperparameters for the warping field to see if this would affect evasiveness, but this did not help. These results indicate that methods designed for evasiveness against input-level detectors do not generalize to being evasive for model-level detectors. Thus, separate approaches are needed for evasiveness against model-level detectors.

|  |  | Acc | Spec | NC | ABS | Param | MNTD |
|---|---|---|---|---|---|---|---|
| Standard | Patch | 53.6 | 63.1 | 65.5 | 52.3 | 46.3 | 59.2 |
|  | Blended | 54.5 | 99.8 | 90.3 | 69.8 | 66.3 | 82.3 |
| TaCT | Patch | 50.8 | 58.3 | 50.9 | 51.6 | 52.7 | 54.4 |
|  | Blended | 50.6 | 78.8 | 68.4 | 61.7 | 64.6 | 94.5 |
| Evasive | Patch | 52.8 | 55.4 | 57.2 | 51.7 | 58.2 | 50.9 |
|  | Blended | 55.6 | 71.2 | 72.8 | 53.8 | 65.3 | 74.4 |
| Evasive+TaCT | Patch | 51.7 | 51.9 | 50.1 | 51.5 | 57.7 | 47.1 |
|  | Blended | 55.7 | 69.3 | 66.0 | 51.0 | 64.5 | 69.6 |

Table 8: Results on source-specific trojans. TaCT obtains highly general evasion, although our evasive trojans are slightly better on average. Combining the two methods yields even greater evasion, demonstrating that TaCT is complimentary with our approach. All values are percent AUROC, and lower is better for the attacker.

**Targeted Contamination Attack (TaCT).**   In our main experiments, we focus on one-to-all attacks. However, one-to-one attacks, also known as source-specific attacks, are an important setting as well. In these attacks, the hidden behavior is only trained to activate on one specific source class. The target class is selected from among the other classes. Tang et al. (2021) find that in this source-specific setting, one can greatly improve evasiveness against Neural Cleanse and MNTD detectors with a simple modification to the standard data-poisoning attack. Instead of just inserting poisoned examples in the source class, they also insert "cover examples", which contain the trigger but are labeled with their original clean label. These cover examples are inserted for all classes besides the source class, which can be considered a form of specificity loss for the source-specific setting. They name this method the Targeted Contamination Attack (TaCT).

TaCT is a method for training evasive trojans in the source-specific setting, and there is some evidence in the original paper that it generalizes across various model-level detectors. To compare our evasive trojans to TaCT, we adapt our standard and evasive trojans for the source-specific setting. This involves only inserting triggers for examples from the source class. For TaCT, we insert cover examples as well. Since TaCT can be combined with our evasive trojans, we include an experiment for this as well. We train 500 trojaned MNIST models for each setting and show results in Table 8.

Interestingly, we find that standard trojans are far harder to detect in the source-specific setting than in the all-to-one setting. Additionally, TaCT greatly improves evasiveness compared to the standard trojans. In fact, it is comparable to our evasive trojans. However, when we combine TaCT with our evasion loss, we obtain the best results. Averaging across all settings, the percent AUROC values for standard trojans, TaCT, evasive trojans, and evasive trojans with TaCT are 66.9, 61.4, 59.9, and 57.2. This shows that TaCT and our evasion loss are complimentary, and in settings where TaCT is applicable we strongly recommend evaluating detectors against it.

|        |         | Standard Trojans | Evasive Trojans |
|--------|---------|------------------|-----------------|
| MNIST  | Patch   | 94.6             | 80.0            |
|        | Blended | 53.9             | 62.6            |
| CIFAR-10 | Patch | 98.8             | 76.9            |
|        | Blended | 99.9             | 100.0           |
| GTSRB  | Patch   | 57.3             | 52.5            |
|        | Blended | 71.3             | 75.4            |
| Average |        | 79.3             | 74.6            |

Table 9: The state-of-the-art PixelBackdoor trojan detector evaluated against standard trojans and our proposed evasive trojans. All values are percent AUROC, and lower is better for the attacker.

## B.3 ADDITIONAL DETECTOR EVALUATIONS

Our six baseline detectors are selected to be representative of various different approaches to model-level detection. Here, we evaluate several additional strong detectors from recent years.

**PixelBackdoor Detector.** Tao et al. (2022) propose an improved optimization process for trigger inversion. Their method builds on the trigger synthesis paradigm introduced by Neural Cleanse, and they obtain stronger detection results when compared to Neural Cleanse. We evaluate their detector on MNIST, CIFAR-10, and GTSRB (due to the long runtime, we did not evaluate on CIFAR-100 for the rebuttal deadline). Results are shown in Table 9. For all the patch attack settings, our evasive trojans are considerably harder to detect. In the blended attack settings our evasive trojans are easier to detect. However, the average AUROC is lower for our evasive trojans. Moreover, when we consider the other detectors in Table 2, the maximum AUROC on blended attacks is still obtained by Neural Cleanse, so the blended results here would not impact the central metric of Max AUROC.

**K-Arm Detector.** Shen et al. (2021) propose the K-Arm detector. This detector iteratively selects the most promising labels for a trigger optimizer and can be thought of as an improved version of Neural Cleanse. On CIFAR-10, standard trojans with patch and blended triggers obtain $52.9\%$ and $89.0\%$ AUROC, respectively. Our evasive trojans obtain $51.1\%$ and $69.5\%$ AUROC, respectively.

