# OpenReview forum: "How Hard is Trojan Detection in DNNs? Fooling Detectors With Evasive Trojans"
_ICLR.cc/2023/Conference — Submitted to ICLR 2023_

### Official Review · Reviewer_jfBX · 2022-10-23

**Confidence:** 4
**Correctness:** 2
**Technical Novelty And Significance:** 2
**Empirical Novelty And Significance:** 2
**Recommendation:** 3

**Clarity, Quality, Novelty And Reproducibility:**

There has been plenty of work on evasive trojans that essentially has the same aim as this paper. Most of these papers propose similar loss functions so it's hard to say the paper is original.

Authors did not share their source code so no reproducibility.

It's generally well written but the claims are weakly supported.

**Strength And Weaknesses:**

+ Specificity is understudied in backdoor attacks and the proposed attack considers to improve specificity.

- Very problematic evaluation and baseline attacks,

**Summary Of The Paper:**

This paper proposes a method to train backdoored models that are stealthier (i.e., they are close to the distribution of clean models). Standard backdoor detection algorithms have less success against these models and they are also more difficult to reverse engineer.

**Summary Of The Review:**

My biggest concern about this paper is the evaluation. First of all, the proposed attack is a supply-chain attack where the attacker has full control over the resulting model. Evasive attacks within this threat model is extensively studied, some examples are [1,2,3]. None of these works are considered in evaluation as baselines. The baseline attacks (Patch - Blend) are vanilla attacks that are shown time and time again easy to detect. Moreover, these baseline attacks are poisoning attacks (not supply chain) which is a significantly more difficult attack to perform. All in all, it is not clear whether this attack brings anything new to the table and whether the claims are true. I would recommend the authors to spend more energy into surveying the SOTA in attacks and defenses, think more carefully about their threat model and perform a more comprehensive evaluation.

Second, there has been more advanced published backdoor defenses the paper has not evaluated. These defenses claim to perform better than old defenses such as NC or ABC. For example: [4,5]

That being said, I like the idea of specifically improving the specificity of the backdoor via loss function. I think this is also a good tool to understand the limits of specificity (e.g., how specific can we make our backdoor, considering the inevitable side effects [6]?)

[1] https://arxiv.org/abs/2002.12200
[2] https://www.usenix.org/conference/usenixsecurity21/presentation/tang-di
[3] https://openaccess.thecvf.com/content/ICCV2021/papers/Li_Invisible_Backdoor_Attack_With_Sample-Specific_Triggers_ICCV_2021_paper.pdf
[4] https://arxiv.org/abs/2102.05123
[5] https://www.cs.purdue.edu/homes/taog/docs/CVPR22_Tao.pdf
[6] https://arxiv.org/abs/2010.09080

---

> ### Author Response · Authors · 2022-11-19
> **Response (1/2)**
>
> Reviewer jfBX,
>
> Thank you for your careful analysis of our work. We hope the following response addresses your concerns.
>
> **Clarifying Our Problem Setting**
>
> We think many of the concerns you mention may come from a misunderstanding of the problem setting that we explore. Our paper is not just proposing a new supply chain attack. Rather, we are investigating whether a certain category of trojan detectors called model-level detectors can be evaded in a generalizable manner. Very few other works have explored this setting, as we explain below.
>
> In trojan detection research, there are several different categories of detectors that correspond to different threat models and use cases. There is general agreement on these distinctions, which have been made in many published works, including [1-3]. In our responses, we use the terminology proposed in [2] of “dataset-level”, “input-level”, and “model-level” detection methods. Dataset-level and input-level detection methods such as Activation Clustering, Spectral Signatures, STRIP, and SentiNet all require inputs with triggers. By contrast, model-level detectors assume that the main source of information we have is the model itself. Detectors in this category include Neural Cleanse, ABS, MNTD, K-Arm, PixelBackdoor, Symmetric Feature Differencing, and many more. Our paper is about training trojans that are evasive for model-level trojan detectors, which is an important setting that very few prior works have explored (see our Related Work section for a full discussion of these prior works).
>
> The distinction between these problem settings is important, because trojans that evade dataset-level and input-level detectors will not necessarily evade model-level detectors (and vice versa). For example, the work of Li et al. [4] and similar papers [5-7] specifically train trojaned models such that inputs with triggers and without triggers are less distinguishable in the input-space or feature-space. While these methods do successfully evade dataset-level and input-level detectors, they are rarely evaluated against model-level detectors (and when they are, only against Neural Cleanse and with whole-image triggers that Neural Cleanse was not designed for). To evaluate whether evasiveness against dataset-level and input-level detectors translates to evasiveness against model-level detectors, we trained 125 CIFAR-10 models with WaNet trojans using different triggers and target labels. Surprisingly, we found that MNTD obtains 100% AUROC on these trojans (in the original WaNet paper, MNTD was not evaluated against). This indicates that different kinds of evasiveness do not necessarily transfer. This is why different works explore different categories of detectors; to the best of our knowledge, we are the first paper to explore general evasiveness against a broad range of model-based detectors.
>
> “Evasive attacks within this threat model is extensively studied, some examples are [1,2,3]. None of these works are considered in evaluation as baselines.”
> To reiterate, we are focused on model-level detection. These papers evaluate on dataset-level and input-level detectors (although see the above caveat about Neural Cleanse). None of them evaluate against the many model-level detectors that have been proposed. Thus, we did not consider them as comparable methods. However, thanks to your suggestion, we included the WaNet experiment described above, which verifies our hypothesis that evasiveness on dataset-level and input-level detectors does not transfer to model-level detectors.
>
>
> “The baseline attacks (Patch - Blend) are vanilla attacks that are shown time and time again easy to detect. Moreover, these baseline attacks are poisoning attacks (not supply chain) which is a significantly more difficult attack to perform.”
>
> This is precisely our point! Model-level trojan detectors have very high performance on existing attacks (including more recent ones like WaNet; see above). Thus, the overarching goal of our paper is to investigate whether we can evade model-level detectors in a generalizable way. We allow any method of insertion, because the method of insertion is not particularly relevant for our core research question.

---

> > ### Author Response · Authors · 2022-11-19
> > **Response (2/2)**
> >
> > **Evaluations on More Recent Detectors**
> >
> > “Second, there has been more advanced published backdoor defenses the paper has not evaluated. These defenses claim to perform better than old defenses such as NC or ABC. For example: [4,5]”
> >
> > We chose our 6 baseline detectors to be representative of a wide variety of different approaches for model-level trojan detection. However, we agree that we could include more recent detection methods. Thanks to your suggestion, we have added experiments for both the K-Arm and PixelBackdoor detectors. The results for the K-Arm detector are that it does not outperform the original baseline detectors in most settings. The results for PixelBackdoor are that it outperforms Neural Cleanse and the other baseline detectors in some settings. We have added these results to the main tables and updated the results section accordingly. The qualitative conclusions remain the same. Thank you for your suggestion. If we have addressed the thrust of your concerns, we kindly ask that you consider raising your score.
> >
> >
> > 1: "Neural Cleanse: Identifying and Mitigating Backdoor Attacks in Neural Networks". Bolun Wang, Yuanshun Yao, Shawn Shan, Huiying Li, Bimal Viswanath, Haitao Zheng, Ben Y. Zhao. IEEE S&P 2019
> >
> > 2: “Detecting AI Trojans Using Meta Neural Analysis”. Xiaojun Xu, Qi Wang, Huichen Li, Nikita Borisov, Carl A. Gunter, Bo Li. IEEE S&P 2021
> >
> > 3: “STRIP: A Defence Against Trojan Attacks on Deep Neural Networks”. Yansong Gao, Chang Xu, Derui Wang, Shiping Chen, Damith C.Ranasinghe, Surya Nepal. ACSAC 19
> >
> > 4: “Invisible Backdoor Attack with Sample-Specific Triggers”. Yuezun Li, Yiming Li, Baoyuan Wu, Longkang Li, Ran He, Siwei Lyu. ICCV 2021
> >
> > 5: "WaNet - Imperceptible Warping-based Backdoor Attack". Tuan Anh Nguyen, Anh Tuan Tran. ICLR 2021
> >
> > 6: "LIRA: Learnable, Imperceptible and Robust Backdoor Attacks". Khoa Doan, Yingjie Lao, Weijie Zhao, Ping Li. ICCV 2021
> >
> > 7: "Backdoor Attack with Imperceptible Input and Latent Modification". Khoa Doan, Yingjie Lao, Ping Li. NeurIPS 2021

---

> > > ### Comment · Reviewer_jfBX · 2022-11-19
> > > **Thanks for the response**
> > >
> > > First, I recommend you look up what supply-chain attack means in the context of the backdoor attack. I don't think I misunderstood your setting.
> > >
> > > Supply-chain attacker trains and provides the model to the defender who may not see the training set, the training objective or other training details. This terminology is defined in one of the very first backdoor attacks against DNNs (https://arxiv.org/pdf/1708.06733.pdf). This is exactly what your paper does, it proposes a new supply-chain backdooring attack that tries to evade existing backdoor-scanner type defenses (such as NC or ABS) or model-level defenses (such MNTD).
> > >
> > > Let's look at the papers I linked and see how their goals are similar to your work:
> > >
> > > 1) Entangled Watermarks as a Defense against Model Extraction: Try to entangle the watermark (which is a just backdoor used defensively) together with the natural latent representations. This aims to make the watermark more robust and harder to detect or remove from the model. They even evaluate their scheme against NC to show that it fails to detect/remove the watermark. They consider a square patch trigger for watermarking (similar to your work). Overall, this is also an evasive supply-chain backdoor attack.
> > >
> > > 2) Demon in the Variant: Statistical Analysis of DNNs for Robust Backdoor Contamination Detection: This is a poisoning-based attack, not a supply-chain one. However, the overall attack goal remains the same: evade backdoor defenses. A supply-chain attacker is strictly stronger than a poisoning attacker so they can apply this attack as well to obtain evasive backdoors. This attack is also evaluated against NC and ABS, which it successfully evades. They consider square patch based triggers (similar to your work). Overall, if you're assuming a strong attacker, you need to realize that any weaker attack is a potential baseline.
> > >
> > > In addition these papers, here's another evasive attack.
> > >
> > > Backdoor Attack with Imperceptible Input and Latent Modification (https://proceedings.neurips.cc/paper/2021/file/9d99197e2ebf03fc388d09f1e94af89b-Paper.pdf) This paper doesn't consider patch-based triggers but their formulation includes a loss term (Equation 4-5) that aims to entangle backdoor and clean latent representations. The impact of this is that backdoor scanner (NC) becomes ineffective. That loss term could easily be used for any other supply-chain backdoor attack to achieve evasiveness.
> > >
> > > All in all, I'm not convinced that not many papers have studied evasive attacks in your setting as you claim. Your formulation is novel but it's not entirely clear that if it does a better job at evading defenses than prior attacks. Judging by Table 2, it looks like detection defenses still have non-negligible success against your attack, although, there's no way to know if these results are any good without any evaluating a prior, evasive backdoor attack as baseline.
> > >
> > > Finally, coming up with a specific threat model doesn't justify rejecting other related work that doesn't fit into this exact threat model. "We are not concerned about backdoor removal defenses", but why? Those defenses are easily applicable in your setting as well, there's nothing inherently preventing a defender from applying NC and Neural Attention Distillation together. So, a threat model that includes model-level defenses that require access to clean data (like MNTD) but not removal defenses is nonsensical, in my opinion.

---

> > > > ### Author Response · Authors · 2022-11-19
> > > > **Response (2/2)**
> > > >
> > > > **Backdoor Attack with Imperceptible Input and Latent Modification**
> > > >
> > > > This attack is among the family of methods that includes WaNet (see the General Response). These methods take an approach of designing triggers and/or optimizing intermediate features on triggered inputs to be inconspicuous compared to clean inputs. Thus, they explicitly focus on making trojans more evasive primarily for dataset-level and input-level detectors. Accordingly, they tend not to evaluate on model-level detectors (except for Neural Cleanse). In the updated paper, we show that WaNet trojans are easily detected by model-level detectors such as MNTD. This demonstrates that evasiveness against dataset-level and input-level detectors does not transfer to model-level detectors. By contrast, our evasive trojans improve evasiveness across a wide range of model-level detectors without specifically training against them.

---

> > > > ### Author Response · Authors · 2022-11-19
> > > > **Response (1/2)**
> > > >
> > > > Hello,
> > > >
> > > > Thank you for your prompt reply and helpful feedback.
> > > >
> > > > **Problem Setting**
> > > >
> > > > We agree that our evasive trojans require access to the training function and thus are a supply-chain attack (see our above response). However, proposing a new supply-chain attack with all possible defenses in mind is not the goal of our paper. Rather, we are focused on the research question of whether generalizable hard-to-detect trojans could be created for model-level detectors; we design our method and experiments solely to answer this research question. This is why we only consider model-level detectors and allow any method of insertion.
> > > >
> > > > "coming up with a specific threat model"
> > > >
> > > > We agree that there are not many papers that focus solely on evading model-level detectors. However, we think this problem setting is well-motivated given the extensive literature on model-level detectors and their unique properties and use cases. The ML community has even developed competitions specifically for model-level trojan detection (e.g., TrojAI), and developing more evasive trojans for this setting would undoubtedly be of interest to this community. We are also not the first paper in this setting: see Sahabandu et al. and Goldwasser et al., who explicitly consider the problem setting of evading model-level detectors. These works are concurrent with our own, and we describe our differences from them in the General Response and Related Work. We are in some sense building on the work of Goldwasser et al. by investigating whether something like undetectable trojans (wrt all existing approaches to model-level detection) could be obtained in DNNs, which is an interesting research question in its own right. To this end, focusing on other categories of defense such as input-level detectors and backdoor removal methods is out of scope (although combining these aims would be an interesting direction for future work).
> > > >
> > > > **Entangled Watermarks**
> > > >
> > > > Many of the results in (Jia et al.) do use small square triggers, similar to the patch triggers that we use. However, this is likely not the case for the Neural Cleanse experiment that they run. The Neural Cleanse results are for (possibly) a single watermarked Fashion-MNIST model. As mentioned on page 9, they sometimes sample "OOD watermarks" from MNIST for Fashion-MNIST models. In Figure 10, which visualizes the Neural Cleanse results, they display an MNIST image as the watermark (i.e., trigger) for Neural Cleanse to reconstruct. Based on this information and the caveat in the Neural Cleanse results section that "watermarked data is not restricted by the degree of perturbation and could even be OOD", we can surmise (with high confidence) that their Neural Cleanse experiment used MNIST images as triggers. Thus, we don't have a strong reason to think that their watermarking approach implicitly tackles model-based trojan detectors any more than the various trojan attacks that *have* been explicitly designed to evade dataset-level and input-level detectors. Since the latter set of methods are more closely related to our work, we focused our efforts on adding experiments on WaNet, which is a representative method from among these papers. Please see the updated paper for our experiments on WaNet.
> > > >
> > > > **Demon in the Variant: Statistical Analysis of DNNs for Robust Backdoor Contamination Detection**
> > > >
> > > > We were not aware of the ABS experiment in (Tang et al.). Thank you for bringing this to our attention. We have added a comparison to their TaCT source-specific trojans in the updated paper. Please see the updated paper for full details. To summarize, their method improves evasiveness across all of our baseline detectors compared to standard trojans. Our evasive trojans perform slightly better on average. However, combining TaCT with our method produces the strongest results, which shows that our methods are complimentary. Thus, we think that TaCT is an excellent evasive trojan method to compare to in the source-specific setting, and we plan to make further updates to our paper to extend this comparison (e.g., on more datasets). Thank you again for bringing this method to our attention. Additionally, please see the updated paper for experiments with the PixelBackdoor and K-Arm detectors that you requested. If we have addressed the thrust of your concerns, we kindly ask that you consider raising your score.

---

> ### Author Response · Authors · 2022-12-13
> **Discussion Reminder**
>
> Thank you again for reviewing our work. We would like to gently remind you that the discussion window is closing soon. We tried our best to address all of your concerns in our responses and revisions. In particular, we added the recommended K-Arm, PixelBackdoor, and TaCT evalations as well as several additional ablations that clarify our contributions. Although the main takeaways of our experiments remain the same, these additional results have strengthened the paper, and we are grateful for your suggestions. We would be happy to hear more from you if you have remaining comments or concerns.

---

### Official Review · Reviewer_zJUx · 2022-10-25

**Confidence:** 3
**Clarity, Quality, Novelty And Reproducibility:** Original work
**Correctness:** 3
**Technical Novelty And Significance:** 3
**Empirical Novelty And Significance:** 2
**Recommendation:** 5

**Strength And Weaknesses:**

Strengths:
1. The author takes the evasion of Trojans as one of the goals and designs a loss function with a clear structure.
2. The author uses 1-Wasserstein distance as a metric, and experiments show that it has a good effect, effectively improving the difficulty of detecting Trojan attacks.
3. The author designed a relatively sufficient experiment, including the performance of the evasion Trojan under different detectors. The results show that the evasion Trojan is harder to detect, its target label is harder to predict, and more difficult to reverse engineer.

Weaknesses:
1. As one of the innovations of this work, the use of 1-Wasserstein distance in the loss function is not elaborated. How the author approximates the infimum also has no further clarification.
2. As the core of the paper, the three-stage loss function designed by the author does not give a theoretical analysis, and the heuristic design of the loss function makes the persuasiveness decrease.
3. I didn't find any text mentioning Figure 1 and Table 2, besides, the experimental results in Table 2 show that the standard Trojan performs even better than the evasion Trojan under the Param detector, which the author does not explain. The figure layout of the entire paper needs to be further optimized.

**Summary Of The Paper:**

This paper uses distribution matching and randomization to reduce Trojan specificity. The author proposes to use 1-Wasserstein distance for the design of the loss function. The authors conduct experiments on MNIST, CIFAR-10 and CIFAR-100, show that the method proposed by the authors can improve the difficulty of Trojans being detected by detectors, and can be applied to various types of Trojans. Experiments also show that the author's proposed method can make Trojans more difficult to reverse engineer.

**Summary Of The Review:**

This work attempts to make Trojans in models harder to detect, which makes sense. I hope the author can elaborate on the design of the loss function in Section 4.1, the use of 1-Wasserstein distance. In addition, the authors need to make a more complete analysis of the experimental results.

---

> ### Author Response · Authors · 2022-11-19
> **Response**
>
> Reviewer zJUx,
>
> Thank you for your careful analysis of our work. We hope the following response addresses your concerns.
>
> **Clarifying Our Use of the W-1 Distance**
>
> "As one of the innovations of this work, the use of 1-Wasserstein distance in the loss function is not elaborated. How the author approximates the infimum also has no further clarification."
>
> As we explain in Section 4.1, the infimum starts off correct at the start of training, because at the start of training the distributional difference is zero (we initialize the trojaned networks as clean networks). As training progresses, we keep the coupling fixed, because if the distances remain small then it will remain a good coupling by definition. We keep the distances small by optimizing the trojaned networks to minimize the distances. This is how we approximate the infimum.
>
> Note that this is not an exact correspondence to the W-1 distance, and it may not even be a very good approximation of the W-1 distance. However, it does provide an intuition for how, in the limit of reducing the distance metrics for each pair of networks, we can approach the distribution of clean networks. Compared to naive alternatives that one might try at first, such as GAN-based approaches, this is far more computationally efficient and still has theoretical grounding. However, it is more appropriate to think of the theoretical grounding in the W-1 distance as an inspiration for our method. We have clarified this in the updated paper.
>
>
> **Theoretical Analysis**
>
> “As the core of the paper, the three-stage loss function designed by the author does not give a theoretical analysis, and the heuristic design of the loss function makes the persuasiveness decrease.”
>
> Requiring theoretical analysis is a highly unusual expectation for neural trojan papers. Many published and highly-cited papers on trojan attacks validate their choices with empirical hypotheses backed up by experiments and ablations. We follow this approach, giving clear reasons for each component of our loss function, which we validate with extensive experiments and ablations.
>
>
> **Other Points**
>
> “I didn't find any text mentioning Figure 1 and Table 2”
>
> This is partly a typo error; our apologies. The link to Table 5 on page 8 should actually link to Table 2. In the updated paper, we have also added a reference to Figure 1. Thank you for pointing this out.
>
>
> “the experimental results in Table 2 show that the standard Trojan performs even better than the evasion Trojan under the Param detector, which the author does not explain.”
>
> In the appendix, we include an extensive ablation that explains why our evasive trojans are particularly susceptible to the Param detector (see Table 6). In particular, we include a discussion of how the Param detector identifies emergent coordination when the randomization loss is removed and how adding the randomization loss fixes this problem. However, even after adding the randomization loss, the Param detector performs slightly better on our evasive trojans compared to standard trojans. This is not a serious issue, because the primary goal of the attacker is to reduce Max AUROC, and on that metric we perform much better than standard trojans. We have clarified this in the updated paper thanks to your suggestion. If we have addressed the thrust of your concerns, we kindly ask that you consider raising your score.

---

> ### Author Response · Authors · 2022-12-13
> **Discussion Reminder**
>
> Thank you again for reviewing our work. We would like to gently remind you that the discussion window is closing soon. We tried our best to address all of your concerns in our responses and revisions, and we would be happy to hear more from you if you have remaining comments or concerns.

---

### Official Review · Reviewer_BFyZ · 2022-10-25

**Confidence:** 4
**Clarity, Quality, Novelty And Reproducibility:** Mentioned in the strength above.
**Correctness:** 3
**Technical Novelty And Significance:** 3
**Empirical Novelty And Significance:** 3
**Recommendation:** 6

**Details Of Ethics Concerns:**

No ethics concerns.

**Strength And Weaknesses:**


B. Strength And Weaknesses

* Strength:
1. The paper is clear and well-written.
2. The novelty is straightforward to understand.
3. Motivation is strong.

* Weaknesses:
1. The analysis is not enough (and not exciting). Some ablation studies are not provided.

**Summary Of The Paper:**

A. Paper summary

- This paper proposes a new trojan method by designing evasive trojans. The key idea is to fine-tune a clean model by leveraging Wasserstein distance to minimize the output logits (between clean output / poisoned outputs) and using l2 distance to regulate the parameter distance (between the clean model and trojan model). Other tricks are provided (e.g., adding random loss) to further reduce the detection rate.

- The result shows it can reduce the detection performance to near-chance levels.


**Summary Of The Review:**

C. Questions:
- In your evaluation, you only show the final detection success rate from different detectors. However, your key motivation is to minimize the output logits/weight difference. Can you show the result that your method is effective in reducing those distances? It would be more interesting if you can show a plot between those distances v.s detection accuracy.

- Following the above questions, why are you specifically choosing the output logits? What if you added the penultimate layer's output?

- In the "randomization" section, you mentioned some empirical results that are not provided in the evaluation. I believe Figure 3 is your motivation; so how about its (i.e., the randomization loss) effectiveness in reducing the detection rate?

I have read many detector papers that are using the distribution of the final layer to detect trojan models. This paper uses Wasserstein distance to minimize this distribution gap between poisoned output logits and clean output logits which is a new challenge to the existing detection methods.

Overall, I think the paper is very clear and highly motivated. However, some ablation studies are missing and I don't think the analysis is exciting. There are a lot of ways to dive deep into the problem and to provide more insights.

Minor: Figure 2 is redundant with Table 2. Your figures and tables are too big.

---

> ### Author Response · Authors · 2022-11-19
> **Response**
>
> Reviewer BFyZ,
>
> Thank you for your careful analysis of our work. We hope the following response addresses your concerns.
>
> **The Evasion Loss Is Correlated With Detection Performance**
> “In your evaluation, you only show the final detection success rate from different detectors. However, your key motivation is to minimize the output logits/weight difference. Can you show the result that your method is effective in reducing those distances? It would be more interesting if you can show a plot between those distances v.s detection accuracy.”
>
> We have added an ablation to the appendix showing that increasing the evasion loss does in fact reduce those distances, and moreover that this corresponds to a steady reduction in the performance of the MNTD detector. Due to time limitations for the rebuttal, we have only included these experiments on MNIST. For the final paper, we plan to add results on the other datasets as well. However, we think the current results (including the various other ablations that we include) provide sufficient evidence that the mechanism behind our approach functions in an understandable manner that matches intuition. Thank you for your suggestion.
>
>
> **Distance Metrics**
> “Following the above questions, why are you specifically choosing the output logits? What if you added the penultimate layer's output?”
>
> We chose the output logits distance because it was one of the first distance metrics that we tried and it worked fairly well. It also directly regularizes the behavior of the network when viewed as a function from inputs to outputs, which is a view taken by many existing detectors, such as Neural Cleanse and MNTD. However, we do include an ablation in the appendix showing that only using the parameter distance metric also results in strong evasion against MNTD. Thus, both distance metrics contribute to our empirical results. We think one could probably design much better distance metrics, although we did not explore this in our work.
>
> We ran an evaluation of the proposed modification to the distance metric where we also add distance in the penultimate feature space. The results are that this improves performance against some detectors and reduces performance against other detectors. This shows that designing effective distance metrics between neural networks for the purpose of training evasive trojans is nontrivial and an interesting direction for future work. We have added these results to the updated paper. Thank you for your suggestion.
>
>
> **Randomization Loss Ablation**
> “In the "randomization" section, you mentioned some empirical results that are not provided in the evaluation. I believe Figure 3 is your motivation; so how about its (i.e., the randomization loss) effectiveness in reducing the detection rate?”
>
> In the appendix, we include an extensive comparison of evasive trojans with and without the randomization loss. These were not included in the main paper due to space constraints. We have made this more clear in the updated paper.
>
>
> **Additional Analysis Experiments**
> We have included additional analysis experiments in the appendix that we think may be interesting to the ICLR audience. These experiments are based off of our anecdotal observations that blended attacks are much easier to detect than patch attacks despite the patch triggers being far more conspicuous (see Figure 2 in the submission for visual confirmation). For our responses to other reviewers, we also included experiments on WaNet [1], a more recent attack that uses nearly invisible warping triggers, and we found that MNTD obtained 100% AUROC (perfect detection) on these networks. This raises the interesting question of whether less visible triggers result in more easily detectable trojans. One possible mechanism for why this would be true is that more conspicuous triggers require fewer modifications to the feature detectors inside of a network in order to link to the desired target label.
>
> If this hypothesis is true, we would expect that decreasing the blending coefficient in our blended trojans would result in larger distances from clean networks in parameter space and easier detection. To test this hypothesis, we ran precisely this experiment by retraining the MNIST blended networks with decreasing blending coefficients. Surprisingly, the results exactly match the prediction: decreasing the blending coefficient for the blended trojans (i.e., making the triggers less visible) leads to a higher distance in parameter-space and easier detection with the Param detector. This is a smooth trend, and we think it provides an interesting potential explanation for why the WaNet models are so easy to detect. We have added these results to the appendix of the updated paper.
>
>
> 1: "WaNet - Imperceptible Warping-based Backdoor Attack". Tuan Anh Nguyen, Anh Tuan Tran. ICLR 2021

---

> ### Author Response · Authors · 2022-12-13
> **Discussion Reminder**
>
> Thank you again for reviewing our work. We would like to gently remind you that the discussion window is closing soon. We tried our best to address all of your concerns in our responses and revisions. In particular, we have run additional ablations showing that reducing our evasion loss leads to lower detectability (the main aspect of the analysis that you may have been concerned about). We would be happy to hear more from you if you have remaining comments or concerns.

---

### Official Review · Reviewer_Wgb5 · 2022-10-29

**Confidence:** 4
**Correctness:** 2
**Technical Novelty And Significance:** 2
**Empirical Novelty And Significance:** 2
**Recommendation:** 5

**Clarity, Quality, Novelty And Reproducibility:**

My comments about the correctness, quality, and novelty are summarized above. No concern about the reproducibility.


**Details Of Ethics Concerns:**

No concern.

**Strength And Weaknesses:**


Strengths:

1. The paper studies an evasive trojan attack that makes existing detection ineffective.
2. The paper conducts experiments with a large number of neural networks.
3. The paper is well-written and easy to read.

Weaknesses:

1. Several prior works explored this idea are missing.
2. The "distribution matching" can introduce several trojan artifacts, leading to detection.
3. The evaluation against backdoor removals is missing.
4. The evaluation against reverse-engineering efforts seems incorrect.


Detailed comments:

This paper studies an evasive trojan attack against existing detection mechanisms. I like the research problem this paper tackles, as most detection techniques proposed so far rely on a specific artifact of backdoors affected by methodologies that the attackers choose.

[Prior work on evasive trojans]

Opposite to the paper's claims (stated in the Introduction), there has also been a vast literature on testing the effectiveness of either detection mechanisms or backdoor removal.

One example is the work by Tan et al. [1]. This work evades existing detection by making the latent representations of clean samples and the samples with the trojan trigger(s) similar. A minor difference between the work by Tan et al. and this paper is whether the loss makes the "latent representations" or the "logits" similar.

Another example is the work by Bagdasaryan et al. [2]. This work formulated the backdoor injection (which is fine-tuning of a clean model, the same as this paper) as multi-task learning for conflicting objectives (which is the superset of the idea proposed by the paper). The work also showed that the attack could evade many existing defenses by varying the objective the attacker adds to the original training objective (L_{task}). Unfortunately, the paper only discusses this work as a proposal for another backdoor attack but misses 60% of the entire paper about evading defenses shown by the authors, which is more valuable.

[1] Tan et al., Bypassing Backdoor Detection Algorithms in Deep Learning, Euro S&P 2020.
[2] Bagdasaryan et al., Blind Backdoors in Deep Learning Models, USENIX Security 2021.


[Concerns about distribution matching]

Distribution matching this paper proposes defines the "distribution" at the input and output spaces of a trojan model. However, the "distribution" can be defined in the latent representation space or feature spaces a network produces while forwarding an input.

(1) This implies that one can detect the trojaned models this attack produces by comparing the latent representations and features. I wonder whether a defender can detect the models in Table 2 with the latent representations. With a fixed set of input examples and several potential trigger patterns, the defender can collect the latent representations from those examples and their versions with the triggers. The defender can run clustering to separate trojaned networks and benign ones.

(2) If evaluation (1) does not offer sufficient detection scores (low AUCs), then I think a defender can easily remove the backdoors from those networks by fine-tuning on a small subset of clean samples for a few epochs (with the same learning rate that we used for training the benign models).


[Weak evaluation]

A line of backdoor defenses that this paper does not consider is backdoor removal. Prior work proposed fine-tuning or fine-pruning as a defense mechanism. They don't need trigger reconstruction or detection; just training (or pruning) a model can reduce the attack success rate.

My second concern is that the evaluation against trigger synthesis or reconstruction is under-studied. NC and ABS are not primarily designed for reconstructing triggers (even if the reconstruction process is important). Some mechanisms specifically study the reconstruction [3]. Due to a large number of parameters, conventional reconstruction leads to potential triggers with no semantic meaning; thus, [3] refines the process to reconstruct more semantically meaningful triggers. Using that advanced reconstruction, the attacker could find the trigger used by an adversary.

[3] Sun et al., Poisoned Classifiers Are Not Only Backdoored, They Are Fundamentally Broken, 2020.

Moreover, the success cases of trigger synthesis are measured by the IoU metric. It's a bit unconvincing that reconstructing the "exact" trigger is necessary to identify backdoored models. One can just extract approximate versions of the original trigger and then exploit them to see whether the classifications are biased or not, which could be sufficient.

**Summary Of The Paper:**

This paper proposes an evasive trojan (backdoor) attack on deep neural networks that makes existing defenses ineffective. The key idea of this attack is "distribution matching." The paper first fine-tunes a network f to construct a backdoored model g that (1) behaves similarly on both clean samples and those with the trigger pattern in the logits and (2) has a minimal difference in their parameter distributions. To increase the specificity of this attack, they further include an objective that the logits of f and g are similar on the same samples with the trigger pattern. In evaluation, the paper demonstrates the attack's effectiveness with a high success rate and a low accuracy drop after backdooring. The attack also evades existing defenses, such as NC, ABS, and MNTD, and the paper shows that the trigger reconstruction was unsuccessful against their backdoored models.

**Summary Of The Review:**

It is important to study the weaknesses of existing defenses from an offensive perspective. From this angle, the paper provides a new attack that the community can use. However, the fact that the prior work has explored similar ideas reduces the novelty of this attack.

In addition, the technical novelty this paper claims (formulating a new objective to produce such evasive trojans) is also studied by several existing works. Unfortunately, this paper misses those works, which gives me concern that the claims are not from a comprehensive review of backdoor attacks and defenses.

Moreover, the idea of "distribution matching" only considers the distributional differences we observe in the output space (the logits), which also gives me concern that the detection can work in the feature space or the latent space. Or, if the attack matches all the distributions, the trojans can be easily removed by just fine-tuning a model.

I like that the evaluation has been conducted with a large number of models (Kudos to the authors), but the evaluation misses several other backdoor defense mechanisms that we even don't need any detection.

I further found that the trigger reconstruction mechanisms have not been evaluated with some advanced techniques or properly, which gives me concern that claiming that "trojan detection is hard" is a bit early.

---

> ### Author Response · Authors · 2022-11-19
> **Response (1/5)**
>
> Reviewer Wgb5,
>
> Thank you for your careful analysis of our work. We hope the following response addresses your concerns.
>
> **Clarifying Our Problem Setting**
>
> We think many of the concerns you mention may come from a misunderstanding of the problem setting that we explore. In trojan detection research, there are several different categories of detectors that correspond to different threat models and use cases. There is general agreement on these distinctions, which have been made in many published works, including [1-3]. In our responses, we use the terminology proposed in [2] of “dataset-level”, “input-level”, and “model-level” detection methods. In particular, dataset-level and input-level detection methods such as Activation Clustering, Spectral Signatures, STRIP, and SentiNet all require inputs with triggers. By contrast, model-level detectors assume that the main source of information we have is the model itself. Detectors in this category include Neural Cleanse, ABS, MNTD, K-Arm, PixelBackdoor, Symmetric Feature Differencing, and many more. Importantly, these methods assume that inputs with triggers are not available. Consequently, some of these methods explicitly search for potential triggers–a task that would become much easier in the other problem settings.
>
> The distinction between these problem settings is important, because trojans that evade dataset-level and input-level detectors will not necessarily evade model-level detectors (and vice versa). For example, the work of Tan et al. [4] and similar papers [5-8] specifically train trojaned models such that inputs with triggers and without triggers are less distinguishable in the input-space or feature-space. While these methods do successfully evade dataset-level and input-level detectors, they are rarely evaluated against model-level detectors (and when they are, only against Neural Cleanse and with whole-image triggers that Neural Cleanse was not designed for). To evaluate whether evasiveness against dataset-level and input-level detectors translates to evasiveness against model-level detectors, we trained 125 CIFAR-10 models with WaNet trojans using different triggers and target labels. Surprisingly, we found that MNTD obtains 100% AUROC on these trojans (in the original WaNet paper, MNTD was not evaluated against). This indicates that different kinds of evasiveness do not necessarily transfer. This is why different works explore different categories of detectors; to the best of our knowledge, we are the first paper to explore general evasiveness against a broad range of model-based detectors.

---

> > ### Author Response · Authors · 2022-11-19
> > **Response (5/5)**
> >
> > 1: "Neural Cleanse: Identifying and Mitigating Backdoor Attacks in Neural Networks". Bolun Wang, Yuanshun Yao, Shawn Shan, Huiying Li, Bimal Viswanath, Haitao Zheng, Ben Y. Zhao. IEEE S&P 2019
> >
> > 2: “Detecting AI Trojans Using Meta Neural Analysis”. Xiaojun Xu, Qi Wang, Huichen Li, Nikita Borisov, Carl A. Gunter, Bo Li. IEEE S&P 2021
> >
> > 3: “STRIP: A Defence Against Trojan Attacks on Deep Neural Networks”. Yansong Gao, Chang Xu, Derui Wang, Shiping Chen, Damith C.Ranasinghe, Surya Nepal. ACSAC 19
> >
> > 4: “Bypassing Backdoor Detection Algorithms in Deep Learning”. Te Juin Lester Tan, Reza Shokri. IEEE S&P 2020
> >
> > 5: "WaNet - Imperceptible Warping-based Backdoor Attack". Tuan Anh Nguyen, Anh Tuan Tran. ICLR 2021
> >
> > 6: “Invisible Backdoor Attack with Sample-Specific Triggers”. Yuezun Li, Yiming Li, Baoyuan Wu, Longkang Li, Ran He, Siwei Lyu. ICCV 2021
> >
> > 7: "LIRA: Learnable, Imperceptible and Robust Backdoor Attacks". Khoa Doan, Yingjie Lao, Weijie Zhao, Ping Li. ICCV 2021
> >
> > 8: "Backdoor Attack with Imperceptible Input and Latent Modification". Khoa Doan, Yingjie Lao, Ping Li. NeurIPS 2021
> >
> > 9: "Topological Detection of Trojaned Neural Networks". Songzhu Zheng, Yikai Zhang, Hubert Wagner, Mayank Goswami, Chao Chen. NeurIPS 2021.

---

> > ### Author Response · Authors · 2022-11-19
> > **Response (4/5)**
> >
> > **Strength of Evaluation**
> >
> > “the evaluation against trigger synthesis or reconstruction is under-studied. NC and ABS are not primarily designed for reconstructing triggers (even if the reconstruction process is important).”
> >
> > While ABS is mainly a detection method, Neural Cleanse is generally considered to be a trigger synthesis method in addition to a detection method. To a point, the NC abstract states “Our techniques identify backdoors and reconstruct possible triggers”, and many subsequent works building on Neural Cleanse such as K-Arm, TABOR, and PixelBackdoor show qualitative trigger synthesis results. Our evaluation of trigger synthesis differs from prior work in that we develop a large-scale quantitative evaluation using thousands of trojaned neural networks. As noted by Qiao et al. [#] and others, standard trojans suffer from the issue of trigger generalization, where inserting one trigger actually inserts multiple triggers. We factor this into our task design by simplifying the task to predicting only the location and shape of the trigger (i.e., the trigger mask). Thus, our trigger synthesis evaluation was carefully designed provide a well-defined quantitative evaluation of trigger synthesis methods.
> >
> > We do agree that there are more trigger synthesis methods that we could include. Thanks to your suggestion, we have added the state-of-the-art PixelBackdoor method from Tao et al. to our baseline methods. This method does not outperform NC in our trigger synthesis task, but it does outperform NC and all of our other baselines in the detection task in some settings (using the negative pattern size as a detection score). In the updated paper, we have added these results to the main tables and modified the results section accordingly. The average AUROC on both standard and evasive trojans increased by a few points, but the qualitative conclusions remain the same.
> >
> >
> > “the success cases of trigger synthesis are measured by the IoU metric. It's a bit unconvincing that reconstructing the "exact" trigger is necessary to identify backdoored models. One can just extract approximate versions of the original trigger and then exploit them to see whether the classifications are biased or not, which could be sufficient.”
> >
> > This may be a misunderstanding. We use the IoU metric precisely because reconstructing the exact trigger is known to be an ill-defined problem. Thus, we simplify the task to predicting the location and shape of a rectangular trigger mask. We agree that reconstructing the exact trigger is usually unnecessary (and in some cases impossible). However, there are some reasons for why one would want to reconstruct semantic properties of the true trigger. Namely, knowing what the trigger is could give one insight into who the attackers might be or what their objectives are. From a technical standpoint, reconstructing the exact trigger it is also an interesting task that many papers have tried to improve performance on. For example, most trigger synthesis papers that include qualitative evaluations of the reconstructed triggers gauge performance precisely based on how qualitatively similar their reconstruction is to the exact trigger. Thus, we do not see this as an unnecessary or uninteresting task. If we have addressed the thrust of your concerns, we kindly ask that you consider raising your score.

---

> > ### Author Response · Authors · 2022-11-19
> > **Response (3/5)**
> >
> > **Blind Backdoors Comparison**
> >
> > “Bagdasaryan et al. … showed that the attack could evade many existing defenses by varying the objective the attacker adds to the original training objective (L_{task}). Unfortunately, the paper only discusses this work as a proposal for another backdoor attack but misses 60% of the entire paper about evading defenses shown by the authors, which is more valuable.”
> >
> > Bagdasaryan et al. evaluate their trojans against exactly two detection defenses: Neural Cleanse and SentiNet. Of these two, only Neural Cleanse is applicable in our setting. As we mention in the related work, their method for evading Neural Cleanse is designed specifically to fool Neural Cleanse and may not generalize to other model-based detectors. Thanks to your suggestion, we have implemented their Neural Cleanse evasion method in our codebase and generated a dataset of 250 trojaned models using their method. Evaluating these models against our baseline detectors reveals that while their methods does improve evasiveness against Neural Cleanse specifically, it does not generalize to other detectors (as we hypothesized). In particular, the Param detector obtains 100% AUROC on their trojans (compared to 70.6% on our evasive trojans in a comparable setting). We have added these results to the appendix of the updated paper.
> >
> > **Backdoor Removal**
> >
> > “(2) If evaluation (1) does not offer sufficient detection scores (low AUCs), then I think a defender can easily remove the backdoors from those networks by fine-tuning on a small subset of clean samples for a few epochs (with the same learning rate that we used for training the benign models).”
> >
> > We do not include backdoor removal defenses, because our central research question is about evading model-level trojan detection, not backdoor removal. We agree that backdoor removal might work quite well on our networks as is, because we are only concerned with inserting trojans that are hard to detect, not hard to remove. We do not see this as reducing the value of our contributions. In fact, we find that our evasive trojans are also harder to reverse-engineer (a surprising finding that we did not expect a priori). Since reverse-engineering is an important step in some backdoor removal proposals (e.g., the unlearning method proposed in the Neural Cleanse paper), the methods that we develop might prove useful for future work on evading backdoor removal, but a comprehensive evaluation on backdoor removal approaches is out of scope for us.

---

> > ### Author Response · Authors · 2022-11-19
> > **Response (2/5)**
> >
> > **Clarifying Our Distribution Matching Loss**
> >
> > “Distribution matching this paper proposes defines the "distribution" at the input and output spaces of a trojan model. However, the "distribution" can be defined in the latent representation space or feature spaces a network produces while forwarding an input.
> >
> > (1) This implies that one can detect the trojaned models this attack produces by comparing the latent representations and features. I wonder whether a defender can detect the models in Table 2 with the latent representations. With a fixed set of input examples and several potential trigger patterns, the defender can collect the latent representations from those examples and their versions with the triggers. The defender can run clustering to separate trojaned networks and benign ones.”
> >
> > We think there may be a misunderstanding here. As we mention in Section 4, our distribution-matching loss “enforces similarity between the distribution of clean networks and trojaned networks”. The distribution is defined over the networks themselves, not merely their input and output spaces. In fact, some of the the baseline detectors explicitly use intermediate features (ABS) or the network parameters themselves (Param). Our notation might have been the source of this confusion; we referred to clean and trojaned models “f” and “g” without explicitly mentioning their parametrization \theta. We have included the parametrization in the updated paper. Thank you for bringing this to our attention.
> >
> > The method that you propose is interesting, and we had planned to implement it to try it out, but some details are underspecified. For instance, if one were to gather intermediate features for multiple inputs and triggers, the dimensionality of the resulting concatenated feature vectors would be many thousands for each network. Since we use training sets of 250 networks for the detection experiments, this would have presented difficulties. One could perhaps get past these difficulties by treating each individual feature vector as a point in a pointcloud or a node in a graph, applying PointNet-style methods or other graph neural networks to classify the collection of features, but this is out of scope for a rebuttal. We do think it would make for an interesting future work! We did actually try a detector similar to this in preliminary experiments–the topological trojan detector proposed by Zheng et al. [9]–but their codebase didn’t work very well on our standard trojans and was quite slow to run, so we left it out.

---

> ### Author Response · Authors · 2022-12-13
> **Discussion Reminder**
>
> Thank you again for reviewing our work. We would like to gently remind you that the discussion window is closing soon. We tried our best to address all of your concerns in our responses and revisions, and we would be happy to hear more from you if you have remaining comments or concerns.

---

### Author Response · Authors · 2022-11-19
**General Response (1/4)**

Reviewers,

Thank you for your careful analysis of our work. We hope the following general response addresses some of your shared concerns and clarifies our contributions. Please see the individual responses as well for a full discussion of specific points.

**Problem Setting and Strength of Evaluation**

In some cases, reviewers may have misplaced our efforts in the broader picture of dealing with the challenges presented by trojans. As noted by reviewers Wgb5 and jfBX, there is significant literature for evading input-level and dataset-level trojan detection, including [1-5]. However, we are solely focused on model-level trojan detection, which is an entirely separate problem setting with a separate literature [6-9]. As we mention in the related work, there have only been a few papers exploring evasive trojans for this important problem setting [10-11], and ours is the first to demonstrate general evasiveness against a broad range of model-based detectors.

Similarly, the concern that our evaluation is insufficient is rooted in the same misunderstanding of what we are trying to do. Methods for evading input-level and dataset-level trojan detection are considerably different from what we explore to the point of being incomparable (however, we do add experiments on these methods thanks to the reviewers’ suggestions; see “Changes to the Paper”). For example, we actually think it is highly likely that some of our evasive trojans would be detected by input-level trojan detectors like STRIP or SentiNet, as we are not concerned with evading this category of detectors. Reviewer Wgb5 also expresses concern that we do not include backdoor removal defenses. We do not include this category of defenses simply because our central research question is about evading model-level trojan detection, not backdoor removal.

We believe that our evaluation does actually answer our central research question of whether generally evasive trojans for model-level detectors can be created. By selecting 6 very distinct baseline detectors, we cover a broad range of different detection approaches. By generating a large-scale dataset containing thousands of clean and trojaned neural networks, we enable precise quantification of detection performance (Note: papers on evading dataset-level and input-level trojans typically only train a handful of models [1-5]). By integrating across all of these evaluation settings, we find clear experimental evidence that our method for making trojans more evasive does in fact work, providing a proof of existence for our central research question.

---

> ### Author Response · Authors · 2022-11-19
> **General Response (4/4)**
>
> 9: “Detecting AI Trojans Using Meta Neural Analysis”. Xiaojun Xu, Qi Wang, Huichen Li, Nikita Borisov, Carl A. Gunter, Bo Li. IEEE S&P 2021
>
> 10: Game of Trojans: A Submodular Byzantine Approach. Dinuka Sahabandu, Arezoo Rajabi, Luyao Niu, Bo Li, Bhaskar Ramasubramanian, Radha Poovendran. arXiv 2022 (concurrent work)
>
> 11: "Planting Undetectable Backdoors in Machine Learning Models". Shafi Goldwasser, Michael P. Kim, Vinod Vaikuntanathan, Or Zamir. arXiv 2022 (concurrent work)
>
> 12: "Poisoned classifiers are not only backdoored, they are fundamentally broken". Mingjie Sun, Siddhant Agarwal, J. Zico Kolter. arXiv 2020
>
> 13: "Blind Backdoors in Deep Learning Models". Eugene Bagdasaryan and Vitaly Shmatikov. USENIX Security 2021
>
> 14: “Better Trigger Inversion Optimization in Backdoor Scanning”. Guanhong Tao, Guangyu Shen, Yingqi Liu, Shengwei An, Qiuling Xu, Shiqing Ma, Pan Li, Xiangyu Zhang. CVPR 2022.
>
> 15: “Backdoor Scanning for Deep Neural Networks through K-Arm Optimization”. Guangyu Shen, Yingqi Liu, Guanhong Tao, Shengwei An, Qiuling Xu, Siyuan Cheng, Shiqing Ma, Xiangyu Zhang. ICML 2021
>
> 16: "Demon in the Variant: Statistical Analysis of DNNs for Robust Backdoor Contamination Detection". Di Tang, XiaoFeng Wang, Haixu Tang, Kehuan Zhang. USENIX Security 2021

---

> ### Author Response · Authors · 2022-11-19
> **General Response (3/4)**
>
> **Clarifying our Contributions**
>
> To clarify our contributions relative to prior work and in light of the reviews, we list our contributions here and give explanations for each one:
> - To the best of our knowledge, we are the first paper to explore whether trojans can be made generally evasive against a broad range of model-level trojan detectors. While many existing works design trojans that are evasive for input-level and dataset-level detectors, there have been very few that explore the important setting of model-level detection. This is perhaps due to the high computational demands of training thousands of clean and trojaned networks, which is only undertaken in model-level trojan detection papers. Related papers that do explore evasiveness against model-level detection include [10, 11, 16], but these papers only evaluate against one or two detectors, or are tied to a specific problem setting. We demonstrate evasiveness against 6 (8 post-rebuttal) detectors, and our method is generally applicable to any DNN classifier and underlying trojan attack.
> - We propose a concrete method for training evasive trojans based on an intuitive W-1 distance interpretation. Compared to naive methods that one might try such as GAN-based methods, this is far more efficient. At the end of the day, our method is conceptually simple, highly effective, and could likely be further improved by developing new distance metrics.
> - We catalog various practical issues that arise when training trojans to be evasive for model-level detectors, including an emergent coordination issue. We propose a randomization loss that fixes this problem.
> - We propose a trojan detector that directly classifies networks based on their parameters (our Param detector baseline). This method mainly helps us identify the emergent coordination issue, but it is also surprisingly performant given that it directly classifies network parameters, which are very high-dimensional objects. Its strong performance is largely due to the summary features that we choose; in preliminary experiments, we found that using the raw parameters resulted in random chance detection performance. To the best of our knowledge, no other detectors like this have been proposed in the literature, and our results show that it possesses unique properties and require additional measures for evasion.
> - We discovered the surprising result that training trojans to be more evasive for model-level detectors also makes them harder to reverse-engineer. To the best of our knowledge, this has not been shown in any prior work, and the security implications are substantial.
> - We developed an experimental methodology for large-scale quantitative evaluations across many detectors and thousands of clean and trojaned networks. To the best of our knowledge, this is the first time an evaluation of this kind has been performed for target label prediction and trigger synthesis (Note: quantitative evaluations have been performed for trigger synthesis before, but only on a small scale). For trigger synthesis, this was nontrivial, because it is well-known that trigger generalization makes trigger synthesis somewhat ill-defined. In light of these results, we recast the problem as predicting only the trigger mask, which is more well-defined, especially when networks are trained to have high specificity.
> - We demonstrated that standard trojans can be detected fairly easily if one has access to the distribution of possible triggers (but not the true trigger). Namely, our Specificity baseline detector outperforms other published detectors on standard trojans despite being very simple. Our proposed specificity loss makes trojans much harder to detect using this method.
>
>
> 1: “Bypassing Backdoor Detection Algorithms in Deep Learning”. Te Juin Lester Tan, Reza Shokri. IEEE S&P 2020
>
> 2: "WaNet - Imperceptible Warping-based Backdoor Attack". Tuan Anh Nguyen, Anh Tuan Tran. ICLR 2021
>
> 3: “Invisible Backdoor Attack with Sample-Specific Triggers”. Yuezun Li, Yiming Li, Baoyuan Wu, Longkang Li, Ran He, Siwei Lyu. ICCV 2021
>
> 4: "LIRA: Learnable, Imperceptible and Robust Backdoor Attacks". Khoa Doan, Yingjie Lao, Weijie Zhao, Ping Li. ICCV 2021
>
> 5: "Backdoor Attack with Imperceptible Input and Latent Modification". Khoa Doan, Yingjie Lao, Ping Li. NeurIPS 2021
>
> 6: "Neural Cleanse: Identifying and Mitigating Backdoor Attacks in Neural Networks". Bolun Wang, Yuanshun Yao, Shawn Shan, Huiying Li, Bimal Viswanath, Haitao Zheng, Ben Y. Zhao. IEEE S&P 2019
>
> 7: "ABS: Scanning Neural Networks for Back-doors by Artificial Brain Stimulation". Yingqi Liu, Wen-Chuan Lee, Guanhong Tao, Shiqing Ma, Yousra Aafer, Xiangyu Zhang. CCS 2019
>
> 8: "TABOR: A Highly Accurate Approach to Inspecting and Restoring Trojan Backdoors in AI Systems". Wenbo Guo, Lun Wang, Xinyu Xing, Min Du, Dawn Song. arXiv 2019

---

> ### Author Response · Authors · 2022-11-19
> **General Response (2/4)**
>
> **Changes to the Paper**
>
> In the updated paper, we have added text clarifying that we are exploring the problem setting of model-level trojan detection and not dataset-level or input-level trojan detection. We have added a section to the related work on methods that are designed to be evasive for dataset-level and input-level detectors (we did not originally cite these methods mainly because we considered them to be a separate subfield of neural trojan research). Thanks to the suggestions of Reviewers Wgb5, BFyZ, and jfBX, we have also added additional comparisons and ablations to the paper and appendix, which we list below.
>
> Relevant to reviewers Wgb5 and jfBX:
> - An evaluation of the Neural Cleanse evasion method from Bagdasaryan et al. [12]. The results are that it does make trojans more evasive against Neural Cleanse specifically, but not against other detectors (as we hypothesize in the Related Work).
> - An evaluation of the more recent and advanced PixelBackdoor trigger synthesis method of Tao et al. [14]. The results are that this outperforms Neural Cleanse and the other baseline detectors in some settings. We have added these results to the main tables and updated the results section accordingly. The average AUROC on both standard and evasive trojans increased by a few points, but the qualitative conclusions remain the same.
> - An evaluation of the K-Arm detector of Shen et al. [15]. The results are that it does not outperform the original baseline detectors in most settings. We have added these results to the main tables and updated the results section accordingly. The quantitative results are not affected very much by this addition, and the qualitative conclusions remain the same.
> - An evaluation of the WaNet attack [2], which is designed to evade dataset-level and input-level detection methods. Surprisingly, we find that WaNet trojans are extremely easy for model-level detectors to identify, with our baselines obtaining 100% AUROC. We believe that this is due to an interesting phenomenon that we anecdotally noticed while running our main evaluations but decided not to include in the paper at the time of submission. Namely, we have observed that trojans that use “invisible” or “stealthy” triggers are actually *easier* to detect for model-level detectors, despite being harder to detect for input-level and dataset-level detectors.
> - While a full evaluation of the phenomenon mentioned in the previous bullet point is out of scope, we have added a small ablation where we vary the blending coefficient in the blended attack. We find that lower blending coefficients (corresponding to more stealthy triggers) actually result in a higher distance in parameter-space from clean networks and easier detection with the Param detector. This is a smooth trend, and it provides a possible explanation for why the WaNet trojans are so easy to detect.
>
> Relevant to reviewer BFyZ:
> - An ablation where we lower the weight on the evasion losses (5 different loss weights; in each case, we train 250 trojaned MNIST networks). Predictably, this increases the distances to the paired clean networks. What’s more, this increase in distance corresponds to an increase in detection performance for MNTD.
> - An evaluation of the proposed modification to the distance metric where we also add distance in the penultimate features space. The results are that this improves performance against some detectors and reduces performance against other detectors. This shows that designing effective distance metrics between neural networks for the purpose of training evasive trojans is nontrivial and an interesting direction for future work.
>
> (Note that due to the deadline, we did not have the chance to make all of these changes. Nearly all of the experiments have been added, and we will add the remaining ones to the final version of the paper.)
> In all cases, these additional comparisons and ablations do not significantly change the qualitative conclusions of our experiments. In most cases, they strengthen our conclusions and provide further evidence of the effectiveness of our evasive trojans.

---

### Decision · Program_Chairs · 2023-01-20

**Decision:**

Reject

**Justification For Why Not Higher Score:**

see above

**Justification For Why Not Lower Score:**

N/A

**Metareview: Summary, Strengths And Weaknesses:**

This work proposed a backdoor attack method with evasive trojans, to evade detection based defenses, utilizing distribution-matching, specificity, and randomization to eliminate distinguishing features of trojaned networks.

There are several important concerns from reviewers:
1. The idea of evasive trojans are not new, as there are several existing works with the similar idea.
2. The evaluation and literature review are inadequate. For example, several advanced attack methods which also controlled the training process are not compared.
3. The design of the proposed loss functions are not well motivated.

Overall, the current shape of this work is not qualified for ICLR.